# Wiring the 'Why': A Unified Taxonomy and Survey of Abductive Reasoning in LLMs

## Abstract

Regardless of its foundational role in human discovery and sense-making, abductive reasoning—the inference of the most plausible explanation for an observation—has been relatively underexplored in Large Language Models (LLMs). Despite the rapid advancement of LLMs, the exploration of abductive reasoning and its diverse facets has thus far been disjointed rather than cohesive. To the best of our knowledge, this paper presents the first survey dedicated specifically to abductive reasoning in LLMs, tracing its trajectory from philosophical foundations to contemporary LLM-based approaches. To address the widespread conceptual confusion and disjointed task definitions prevalent in the field, we establish a unified two-stage definition that formally categorizes prior work. This definition disentangles abduction into *Hypothesis Generation*, where models bridge epistemic gaps to produce candidate explanations, and *Hypothesis Selection*, where the generated candidates are evaluated and the most plausible explanation is chosen. Building upon this foundation, we present a comprehensive taxonomy of the literature, categorizing prior work based on their abductive tasks, datasets, underlying methodologies, and evaluation strategies. In order to ground our framework empirically, we conduct a compact benchmark study of current LLMs on abductive tasks, together with targeted comparative analyses across task stage (generation vs. selection), model sizes and families, metric choices, and dataset/task characteristics such as domain, context length, and output structure. Moreover, by synthesizing recent empirical results, we examine how LLM performance on abductive reasoning relates to deductive and inductive tasks, providing insights into their broader reasoning capabilities. Our analysis reveals critical gaps in current approaches—from static benchmark design and narrow domain coverage to narrow training frameworks and limited mechanistic understanding of abductive processes. Finally, we propose research directions spanning richer evaluation frameworks, reinforcement learning for explanatory virtues, multi-agentic architectures, and circuit-level interpretability to advance the field toward more rigorous abductive reasoning capabilities.

## 1 Introduction

A doctor faces a patient whose symptoms do not quite fit any textbook disease. A physicist sees a measurement that should have been impossible. A detective walks into a room where the clues do not add up. In each case, the crucial step is not merely applying known rules, but *guessing* what might be going on behind the scenes: inventing a plausible story that, if true, would make the puzzling observation unsurprising.

This is **abductive reasoning**: given an observation $O$, infer a hypothesis $H$ such that, if $H$ were true, $O$ would make sense (Peirce, 1931–1958, 5.189). A simple example is everyday troubleshooting: your browser shows an error, and you infer that "the Wi-Fi might be down," because if that hypothesis were true, the error would be expected. First formalized by philosopher Charles Sanders Peirce, abduction is the cornerstone of human sense-making, scientific discovery, and diagnostic reasoning (Aliseda, 2006, p. 28). It is the ampliative, fallible leap from data to explanation (Paul, 1993; Douven, 2025).

In recent years, the pursuit of artificial general intelligence has rekindled immense interest in computationally modeling abduction. As shown in Figure 1, the rise of Large Language Models (LLMs) has led to an explosion

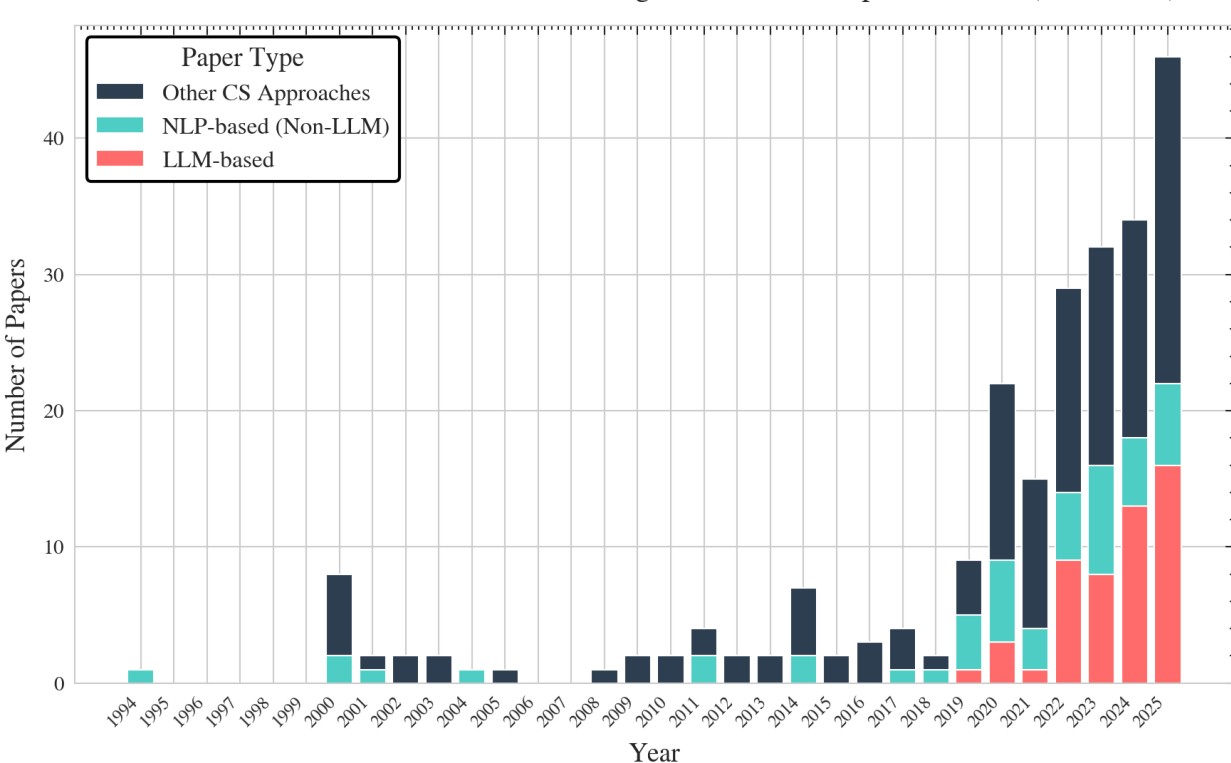

Figure 1: **Publication Trends of Abductive Reasoning Research in Computer Science.** The stacked bar chart categorizes publications into LLM-based, NLP-based (Non-LLM), and other computer science approaches.

of research aimed at equipping these models with abductive capabilities. Early efforts in NLP focused on commonsense reasoning tasks, such as generating explanations for everyday situations or selecting the most likely cause from a pair of options (Bhagavatula et al., 2020; Du et al., 2022; Zhao et al., 2024). More recent work has expanded into a variety of diverse and complex domains. For instance, researchers have explored multimodal contexts that combine text with vision to ground abductive inferences (Hessel et al., 2022; Liang et al., 2022; Zhao et al., 2022; Li et al., 2023). Other studies have investigated programmatic rule learning to infer underlying logic from observed patterns (Kim et al., 2022). In specialized professional fields, abductive reasoning models are increasingly being applied to assist with complex medical diagnosis (Tchango et al., 2022; Wu et al., 2025b) and legal case analysis (Nguyen et al., 2023). Furthermore, these methods have been integrated with knowledge graphs to predict missing relationships and derive structural explanations (Bai et al., 2024; Du et al., 2019). Finally, abductive capabilities are even being extended into highly rigorous domains, including formal logic and rule-based reasoning (Young et al., 2022; Tafjord et al., 2021; He & Lu, 2024; Sheng et al., 2025). Despite its importance for explanation, diagnosis, and discovery, abductive reasoning has historically received much less attention than deduction in AI and NLP. While deductive reasoning has been supported for years by well-established textual entailment and natural language inference benchmarks, abductive reasoning has only more recently begun to accumulate dedicated datasets and task formulations (Giampiccolo et al., 2007; MacCartney & Manning, 2007; Bowman et al., 2015; Williams et al., 2018; Camburu et al., 2018; Bhagavatula et al., 2020).

Despite this surge in research, the field of abductive reasoning in AI is heavily fragmented. There is no consensus on a unified definition or a standard operationalization of the task. Some research frames abduction purely as a discriminative task of *selecting* or ranking the best hypothesis from a predefined set (Bhagavatula

et al., 2020; Tchango et al., 2022; Chan et al., 2023), or evaluating the plausibility of a single candidate hypothesis (He & Lu, 2024; Rudinger et al., 2020). Other work treats it as a purely generative task of *creating* a free-text explanation from scratch (Bhagavatula et al., 2020; Zhao et al., 2024), or generating missing structured knowledge such as facts and programs (Tafjord et al., 2021; Young et al., 2022). Yet others explore structured hypothesis spaces like logical rules or knowledge graph relations (Bai et al., 2024; Gao et al., 2026b). This lack of a common framework makes it challenging to compare methodologies, evaluate progress cumulatively, and identify critical avenues for future research.

To address this fragmentation, we first investigate the philosophical and historical foundations of abductive reasoning—tracing its evolution from early precursors and Peirce's foundational work into modern epistemology. This historical grounding steers us to propose a unified working definition, drawing closely from the concept of Inference to the Best Explanation (IBE) (Harman, 1965; Lipton, 2004), that organizes and unifies the field. Rather than viewing abduction as a monolith, we conceptualize it as a **two-stage process**, detailed in Section 2.2.1. In the first stage, **Hypothesis Generation**, the goal is to produce a creative and ampliative set of candidate hypotheses $\{h_1, h_2, \ldots, h_n\}$ that could potentially explain the observation. In the second stage, **Hypothesis Selection**, these candidates may be evaluated—among other possible approaches—in terms of explanatory virtues such as simplicity, coherence, and predictive power, in order to identify the best available explanation, $h^*$.

This two-stage model allows us to organize the vast and varied literature into a coherent structure. (Note that this survey covers research published through February 2026.) Using this lens, in Section 3, we develop a comprehensive four-axis taxonomy to categorize existing work based on **Task Formulation** (Generation, Selection, or Full Pipeline), **Dataset Type** (commonsense vs. expert/formal, textual vs. multimodal), **Methodology** (prompting, fine-tuning, retrieval-augmented, multi-agent, hybrid/neuro-symbolic), and **Evaluation Approach** (accuracy-based, non-accuracy-based, human evaluation).

Beyond theoretical classification, this survey provides a compact empirical analysis (Section 4). We benchmark a diverse array of modern LLMs—spanning different model families and scales from 3B to 72B parameters (including Qwen2.5, Qwen3, Llama3.1, Llama3.3, DeepSeek-V3.2, GPT-4o, and GPT-5.4)—across representative tasks for both generation and selection. Using a spectrum of datasets from commonsense to expert formal domains (ART, e-CARE, UNcommonsense, DDXPlus, ProofWriter, AbductionRules, NeuLR, True Detective, and MuSR), our benchmarking and targeted follow-up studies reveal the nuanced strengths and limitations of current models under varying levels of abductive complexity. Furthermore, by aggregating published results from existing studies, we provide a comparative empirical analysis of LLM capabilities across abductive, deductive, and inductive reasoning paradigms.

Finally, synthesizing our findings, we pinpoint the most pressing gaps limiting the field: conceptual fragmentation, narrow domain coverage, structurally shallow benchmarks, and a disconnect between benchmark accuracy and genuine abductive reasoning. Current supervised training paradigms and static, single-shot benchmarks fail to capture how abductive reasoning is embedded in richer interactive multi-step settings, and higher accuracy does not necessarily imply genuine explanatory inference. To overcome these limitations, we suggest several future directions: reinforcement learning with rewards aligned to broader explanatory virtues, richer action-oriented benchmarks grounded in expert domains, systematic exploration of abduction across fields such as medicine, law, and engineering, generalized multi-agent architectures that decompose the abductive pipeline across role-specialized agents, and advancing mechanistic interpretability to uncover the circuit-level mechanisms that distinguish true abduction from other forms of logical inference.

Several recent surveys partially overlap with our topic, but they take broader or adjacent perspectives. Luo et al. (2023) and Yang et al. (2023) survey logical reasoning more broadly, covering deduction, induction, and abduction. He & Chen (2025) survey hypothesis discovery and rule learning with LLMs, overlapping most clearly with our discussion of abductive generation and scientific discovery. By contrast, we focus specifically on abductive reasoning in LLMs, introducing a unified two-stage definition that separates hypothesis generation from hypothesis selection, an abductive-specific four-axis taxonomy, and a compact benchmark spanning both stages. We therefore position this work, to the best of our knowledge, as the first survey dedicated specifically to abductive reasoning in LLMs.

**Contributions**

The primary contributions of this survey are as follows:

- **A Unified Definitional Framework:** By examining the philosophical foundations of abduction and IBE, we formalize a unified two-stage (generation and selection) definition. This framework resolves existing conceptual fragmentation and organizes the literature.

- **A Comprehensive Taxonomy and Structured Literature Review:** We introduce a novel, four-axis taxonomy that categorizes the landscape of abductive reasoning research across task formulations, dataset domains and modalities, methodologies, and evaluation paradigms. Utilizing this framework, we provide an extensive review of modern abductive reasoning research, situating more than 60 key papers to highlight trends, relationships, and research gaps.

- **Empirical Benchmarking and Analysis:** We present a novel empirical evaluation of modern LLMs (spanning 3B to 72B parameters, including both open-weights and proprietary models) across diverse commonsense and expert domains, covering both Stage I (generation) and Stage II (selection) tasks. Additionally, by aggregating published results from existing literature, we conduct a cross-paradigm comparative analysis to provide valuable insights regarding the relationship between deductive, inductive, and abductive reasoning paradigms.

- **Key Directions for Future Work:** We identify critical gaps in the current literature—such as structurally shallow benchmarks, narrow domain coverage, over-reliance on static benchmarks, and limited transparency. To address these, we outline strategic future directions, emphasizing RL-based optimization for explanatory virtues, the development of richer action-oriented and cross-domain benchmarks, the design of generalized multi-agentic abductive frameworks, and advancing the mechanistic interpretability of neural abductive circuits.

**Guide for the reader.** Because the literature uses "abductive reasoning" in inconsistent ways, we begin in Section 2.1 with the philosophical background that explains this fragmentation and motivates our two-stage framework. Readers mainly interested in the survey's operational framework can begin with Section 2.2, which states the working definition that anchors the rest of the paper, and then continue to Section 3 and Section 4, returning to Section 2.1 as needed. We retain the historical discussion because it is part of the argument for the framework rather than only background context.

# 2 Background and Definitions

## 2.1 Philosophical and Historical Foundations

The current landscape of abductive reasoning research in Large Language Models is characterized by a substantial degree of definitional fragmentation. Studies frequently invoke the term "abductive reasoning" to describe distinct tasks, adopt divergent theoretical stances, and operationalize the concept through disparate computational formulations. Some work frames abduction as a ranking problem over plausible hypotheses (Zhu et al., 2020); others ground it in set-cover formalism within claim verification pipelines (Dougrez-Lewis et al., 2025); still others equate it with Inference to the Best Explanation, evaluating explanations via criteria such as parsimony and coherence (Dalal et al., 2024). This widespread ambiguity, however, is not a modern artifact of computational expediency; it is largely an inheritance of a philosophical debate that has unfolded over more than a century. Understanding this historical trajectory is therefore not a peripheral scholarly exercise but rather a crucial step for diagnosing the underlying causes of the field's current inconsistency and for constructing the unified framework that this survey proposes. Moreover, the two-stage pipeline we adopt as our working definition is a conceptual architecture that best becomes well-motivated and clearly interpretable when one traces its lineage.

Our historical and conceptual discussion is deeply informed by Aliseda's work on the logic of abduction (Aliseda, 2006) and Dellsén's examination of abductive reasoning in scientific contexts (Dellsén, 2024).

Rather than engaging with the distinct philosophical theses these authors advance, we rely on their comprehensive historical syntheses to map the evolution of the concept. These works provide the essential philosophical grounding for this section, which we have adapted and reoriented to address the specific contexts and challenges of LLM research, allowing us to trace the roots of this definitional fragmentation.

### 2.1.1 Precursors and Early Philosophical Roots

Long before abductive reasoning received a formal name, its core intuition—explaining observations by hypothesizing their underlying causes—was already present in scientific and philosophical practice. Aristotle's concept of *apagoge*, a form of reasoning that yields conclusions that are merely probable rather than certain, is widely considered the earliest ancestor of what we now call abduction.

For centuries after, philosophers and scientists pursued what was termed a "Logic of Discovery": a rigorous method that could reliably generate new, guaranteed knowledge. Thinkers such as Francis Bacon, René Descartes, and Gottfried Wilhelm Leibniz each proposed strict methodologies—rooted in induction or deduction—to achieve this goal. In practice, however, their actual scientific work told a different story. These figures, along with later scientists like Antoine Lavoisier and Charles Darwin, routinely proposed theories not because they were logically guaranteed, but because they offered the best available explanation for observed phenomena. In other words, they were implicitly performing abductive reasoning.

By the mid-19th century, the tension between the pursuit of infallible discovery methods and the reality of this hypothesis-driven scientific practice led to a significant philosophical shift. Scholars increasingly accepted *fallibilism*—the view that knowledge claims need not rest on absolute certainty—and began to formally separate the *generation* of scientific hypotheses from their subsequent *justification*. This distinction between how explanations are created and how they are evaluated is a recurring theme throughout the history of abductive reasoning, and as we will see in Section 2.2.1, it forms the structural backbone of our proposed framework for analyzing abduction in LLMs. Additionally, this philosophical shift not only resolved the gap between theoretical logic and applied scientific practice, but it also set the stage for Peirce to formally categorize this probabilistic, explanatory "guessing" as a distinct logical operation.

### 2.1.2 The Founding Father: Charles S. Peirce

The American pragmatist **Charles S. Peirce** (1839–1914) is recognized as the philosopher who gave abduction its logical form. While he used terms like "Hypothesis" and "Presumption" in earlier works (Peirce, 1931–1958, 2.776, 2.623), "Retroduction" and "Abduction" (Peirce, 1931–1958, 1.68, 2.776, 7.97) became his standard terminology for this mode of inference.

**Peirce's Logical Formulation**  Peirce contrasted Abduction with Deduction and Induction, positioning abduction as the first stage of logical inquiry. He provided the following famous schema (Peirce, 1931–1958, 5.189):

> *The surprising fact, C, is observed.*
> *But if A were true, C would be a matter of course.*
> *Hence, there is reason to suspect that A is true.*

According to Peirce, Deduction proves that something *must* be; Induction shows that something *actually is operative*; but **Abduction merely suggests that something *may* be** (Peirce, 1931–1958, 5.171). Crucially, Peirce viewed abduction as "the only logical operation which introduces any new ideas" (Peirce, 1931–1958, 5.171).

**Generation, Selection, and the Ambiguity in Peirce's Account**  Peirce's exact definition of abduction is notoriously difficult to pin down, as his views on the matter evolved over his long career and arguably lacked strict consistency. A central point of contention revolves around the scope of the abductive act: whether it encompasses solely the creation (or generation) of hypotheses that can account for observations, or if it also includes the subsequent selection of the best, most plausible, or most probable explanation among

competing alternatives. Peirce himself contributed to this confusion by describing abduction both as the "process of forming an explanatory hypothesis" (Peirce, 1931–1958, 5.171) and as the "process of choosing a hypothesis" (Peirce, 1931–1958, 7.219). This apparent ambiguity has led some scholars to conclude that Peirce held no coherent or unified view on abduction at all (Frankfurt, 1958).

However, the predominant interpretation within philosophical and logical academia is that Peirce's most influential formulations isolate abduction to the first stage: the psychological and logical process of generating or suggesting new hypotheses (Hanson, 1958; Kapitan, 1992; Minnameier, 2004; Campos, 2011). Under this standard reading, Peircean abduction primarily describes the mechanism by which we come to think of novel theories that could potentially explain the facts before us, regardless of whether those theories can immediately be considered true or highly plausible. This stands in contrast to contemporary epistemological views, which generally adopt a broader framework where abductive reasoning—commonly equated with inference to the best explanation—intrinsically involves both the generation of candidates and the epistemic process of evaluating and selecting the most probable explanation (Lipton, 2008; Douven, 2025; Campos, 2011; Jiang, 2024).

Crucially, this historical terminological disagreement has transmitted directly into research on abductive reasoning in AI and large language models (LLMs). By explicitly acknowledging this deep-seated definitional split, one of the main goals of this survey is to establish a grounded, pragmatic consensus within our field.

### 2.1.3 Modern Developments: Inference to the Best Explanation

The modern concept of Inference to the Best Explanation (IBE) has become essentially synonymous with our contemporary understanding of abductive reasoning in epistemology and philosophy of science. IBE characterizes the process by which we reason from observed phenomena to their most satisfactory explanations, accepting hypotheses not merely because they account for the data, but because they do so better than available alternatives. This equivalence between IBE and abductive reasoning—though not without nuance—justifies treating developments in IBE theory as directly relevant to our understanding of abduction. Because of this fundamental connection, tracing key developments in IBE theory illuminates the evolution of abductive reasoning itself.

**Harman's conception of IBE**  Gilbert Harman's seminal 1965 paper introduced IBE as a distinct form of non-deductive inference. Harman characterized it as inferring a hypothesis from the premise that it would provide a better explanation for the evidence than any competing hypothesis (Harman, 1965). This formulation marked a significant departure from earlier accounts by making the inference explicitly comparative—one accepts a theory not simply because it explains the data, but because it explains the data better than alternatives. Harman acknowledged that judging which hypothesis is "sufficiently better" depends on considerations such as simplicity, plausibility, explanatory scope, and freedom from ad hoc elements, though he declined to elaborate on the precise nature of these criteria. Importantly, Harman's account focuses primarily on the evaluative and selective dimension of explanatory reasoning—the second stage in the broader two-stage conception of abductive reasoning we discussed earlier. While Peirce emphasized the creative generation of explanatory hypotheses, Harman's IBE concentrates on the epistemic justification for accepting one hypothesis over others once candidates have been identified.

**Lipton's two-stage framework**  Peter Lipton's influential work refined and expanded Harman's framework in ways that prove particularly significant for our purposes (Lipton, 2004). Lipton explicitly articulated IBE as a two-stage process: first, a generation stage in which a limited set of plausible competing hypotheses is produced (resonating with Peirce's emphasis on hypothesis formation); second, an inference stage in which the best among these candidates is selected and accepted based on explanatory considerations. This bifurcated structure provides a more complete account of abductive reasoning by acknowledging both the creative and evaluative dimensions. Lipton's framework is especially significant for our investigation because it provides the conceptual foundation upon which we will build our own characterization of abductive reasoning in LLMs.

## 2.2 A Unified Framework and Related Concepts

### 2.2.1 A Working Definition

One of the significant challenges to evaluating abductive reasoning in AI, particularly within Large Language Models, is the general absence of a shared, rigorous definition. This ambiguity stems from a historically contentious discourse among logicians regarding the precise meaning of abduction. Consequently, the translation of "abductive reasoning" into computational tasks remains highly variable and fragmented. Researchers often structure the concept to align with their specific objectives. For example, some frameworks focus entirely on the open-ended generation of plausible hypotheses (He et al., 2025a), whereas others operationalize it merely as selecting the best explanation from predefined options (Chan et al., 2023). Because the term is broadly applied to such diverse methods, making direct comparisons between studies is particularly challenging. Naturally, this methodological fragmentation has yielded mixed empirical results regarding model capabilities. Some surveys suggest that abductive reasoning is the easiest of the three primary types of logical reasoning for language models (Luo et al., 2023). In contrast, other assessments reach different conclusions, finding that models often struggle with abduction despite demonstrating strong proficiency in deductive reasoning (Dougrez-Lewis et al., 2025).

A fundamental goal of this survey is to establish a practical working definition of abductive reasoning. Our intention is not to resolve the long-standing debates within logic and philosophy, but rather to introduce a functional framework tailored to the needs of the LLM research community. This definition, informed by the historical context, is intended to unify our fragmented field, enabling researchers to precisely situate their contributions and compare results on a common basis. We subsequently employ this framework to compare and categorize the existing literature.

To encompass the diverse array of approaches in current research, we adopt a framework closely aligned with Peter Lipton's view of Inference to the Best Explanation (IBE) (Lipton, 2004) (as discussed in Section 2.1.3). This perspective, which has been explored in recent work on System 2 reasoning pipelines for LLMs (Zheng et al., 2025) and in interpretable evaluation frameworks grounded in IBE (Dalal et al., 2024), effectively encompasses almost all current work within the field, as demonstrated in Table 2, which organizes prior studies. We define abductive reasoning as a **two-stage process** aimed at transforming a set of observations into a plausible explanation. This process can be viewed as a functional pipeline (illustrated in Figure 2):

1. **The Observation (The Stimulus):** The process is initiated by an observation or a set of facts ($O$). While abductive reasoning can be applied to mundane or trivial occurrences, its primary epistemic value—and its historical root in Peircean logic (Peirce, 1931–1958, 5.189)—lies in addressing observations that are "surprising," "anomalous," or "non-obvious" given the agent's prior knowledge (background theory $T$). The goal is to resolve the epistemic gap presented by $O$.

2. **Stage I: Hypothesis Generation (The Creative/Retrieval Phase):** In the first stage, the reasoner generates a set of candidate hypotheses $\mathcal{H} = \{h_1, h_2, ..., h_n\}$, a process ranging from the retrieval of known schemas (recognizing a familiar pattern in prior knowledge) to genuine creative discovery (synthesizing entirely new theories, as in scientific discovery).

3. **Stage II: Hypothesis Selection (The Evaluative Phase):** In the second stage, the reasoner evaluates the candidate set $\mathcal{H}$ to identify the "best" explanation $h^*$ (or a ranked subset). For the *mechanism*, this could involve pruning, ranking, and distinguishing between competing hypotheses. Regarding the *criterion*, selection might be guided by explanatory virtues such as simplicity (Occam's razor), consistency, coherence with background knowledge, and predictive power.

By adopting this two-stage definition of abductive reasoning, we establish a common ground that acts as a union of the LLM literature focused on the open-ended generation of possibilities and works restricted to discriminating among given hypotheses, ensuring that future endeavors and past efforts are cohesively categorized.

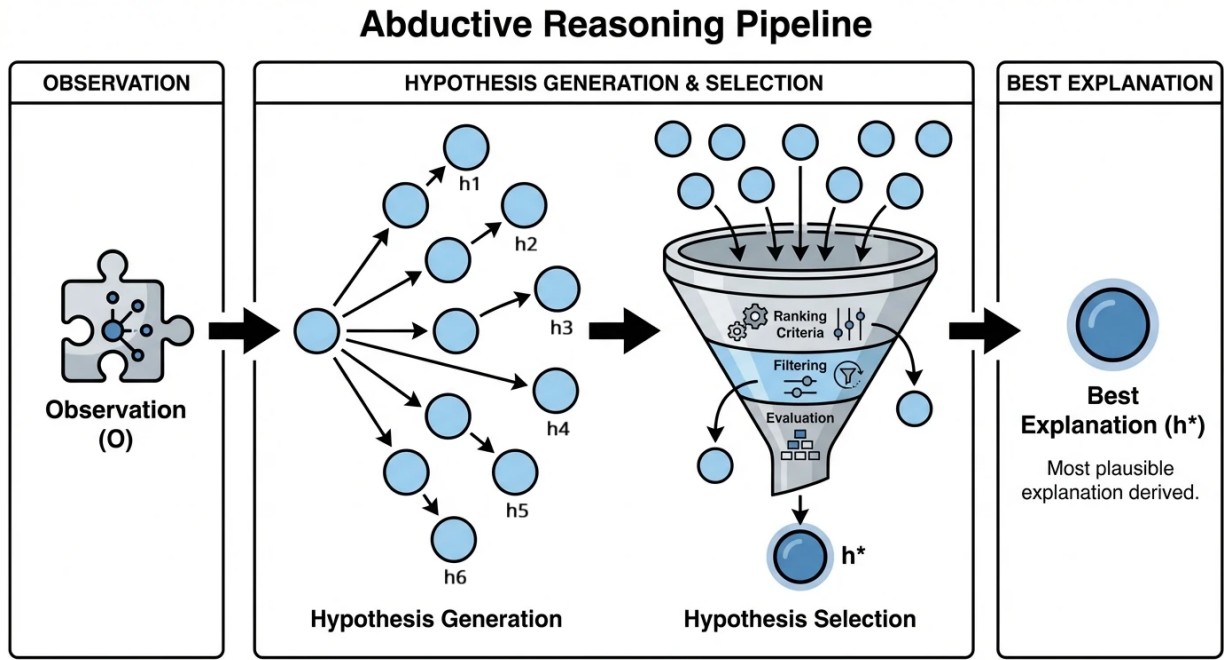

Figure 2: **The Abductive Reasoning Pipeline.** As defined in Section 2.2.1, we conceptualize abduction as a two-stage process: (1) **Hypothesis Generation**, where an observation ($O$) triggers the retrieval or creation of a candidate set $\mathcal{H}$ to bridge an epistemic gap; and (2) **Hypothesis Selection**, where candidates are evaluated to identify the best explanation $h^*$. This survey unites works that focus on either stage individually under this common framework.

### 2.2.2 Relationships with Other Reasoning Paradigms

Alongside establishing a unified working definition, it is essential to contextualize abductive reasoning by examining how it relates to adjacent modes of inference. Illuminating these conceptual intersections and theoretical boundaries is critically important, as clarifying how abduction interacts with related paradigms can foster cross-pollination among implicitly linked research ecosystems and substantially enrich progress in the field.

**Positioning Abduction within Logical Taxonomies.** Abductive reasoning does not exist in an isolated vacuum; it is situated within three fundamental, highly interrelated spectra of logical reasoning. While often used interchangeably in computational literature, these three paradigms highlight different facets of reasoning under uncertainty:

- **Defeasible vs. Non-defeasible (The Epistemic Status):** This spectrum focuses on the permanence of a conclusion. Unlike non-defeasible logic, where truth is absolute once established, *defeasible reasoning* supports plausible or tentative conclusions that might be retracted or overturned if new, contradicting information becomes accessible.

- **Monotonic vs. Non-monotonic (The Structural Mechanics):** This spectrum focuses on the formal accumulation of knowledge. Classical reasoning is monotonic because adding new premises strictly preserves previously valid arguments ($A \vdash B \implies A, C \vdash B$). *Non-monotonic frameworks*, however, allow the inclusion of new premises to formally invalidate former conclusions. Defeasibility and non-monotonicity are practically two sides of the same coin: defeasibility is the conceptual property, while non-monotonicity is the formal logical mechanism that permits it.

- **Ampliative vs. Non-ampliative (The Semantic Content):** This spectrum contrasts the information yielded by an inference. *Ampliative logic* pushes beyond the mere information contained strictly within the premises, generating intrinsically new insights. Non-ampliative logic only reformulates existing data.

Abductive reasoning inherently operates as a ***defeasible***, ***non-monotonic***, and ***ampliative*** process (as summarized in Table 1) (Paul, 1993; Delrieux, 2004; Douven, 2025). Because these three concepts share deep conceptual and structural similarities when handling uncertainty, they form a highly interconnected research ecosystem. Abduction sits at the convergence of these traits. As a consequence of this overlap, the challenges and solutions associated with modeling these modes of reasoning are broadly transferable. Therefore, methodological breakthroughs in the field of Large Language Models (LLMs) for defeasible reasoning, non-monotonic logic, and ampliative inference can naturally inform and substantially enrich the ongoing development of abductive reasoning capabilities in LLMs.

Table 1: Comparison of the three classical reasoning paradigms across key logical properties.

| Property | Deduction | Induction | Abduction |
|---|---|---|---|
| *Defeasibility* | Non-defeasible | Defeasible | Defeasible |
| *Monotonicity* | Monotonic | Non-monotonic | Non-monotonic |
| *Ampliativeness* | Non-ampliative | Ampliative | Ampliative |

**Abduction vs. Deduction.** The contrast between abduction and deduction is the most straightforward one. Deductive reasoning represents the paradigm of absolute necessity and mathematical certainty (Peirce, 1931–1958, 4.145). It operates by guaranteeing conclusions from premises in a truth-preserving manner: given true premises and valid inference rules, the conclusion is necessarily true. This makes deduction non-defeasible, monotonic, and non-ampliative—the polar opposite of abduction on every spectrum (Table 1). Despite this fundamental divergence, the two interact closely in practice: deduction is occasionally employed *within* the abductive pipeline itself—for instance, to verify whether a candidate hypothesis logically entails the observed data during hypothesis selection, or to derive testable predictions from abduced explanations. (Zheng et al., 2025; Shanahan, 1989; Piazza et al., 2023)

**Abduction vs. Induction.** The relationship between abduction and induction is highly contentious and represents a longstanding source of philosophical debate (Flach & Kakas, 2000; Bessant, 2000). Some scholars go so far as to question whether there even exist "absolute, Platonic ideals of abduction and induction" that could settle the matter, whether "induction and abduction are conceptually distinct modes of reasoning" at all, and whether "they can be modelled computationally in the same, or similar, ways" (Psillos, 2000). As Table 1 illustrates, induction and abduction share an *identical* profile across all three logical dimensions—both are generally defeasible, non-monotonic, and ampliative.

Many traditional perspectives classify both as mere subsets of a single "non-deductive reasoning" framework; the broad understanding of induction in computational philosophy, for instance, often subsumes abduction entirely by defining induction as any inference that expands knowledge under uncertainty (Thagard, 1988). Similarly, the influence of Mill (Mill, 1858) led many to characterize all methods of generating causal hypotheses under the umbrella term of "induction." Conversely, others argue that induction is merely a specific instance of abduction—particularly when abduction is understood comprehensively as Inference to the Best Explanation, where enumerative induction becomes a special case (Harman, 1965). Because these borders are so thoroughly blurred, it is vital that research communities working in either domain proactively inform one another.

# 3 Categorization of Literature

## 3.1 By Task Formulation

Abductive reasoning tasks differ mainly in how they operationalize the underlying inference problem. Following our two-stage working definition (Section 2.2.1), we group them by the component of the abductive process they directly instantiate: Stage I, *hypothesis generation*, and Stage II, *hypothesis evaluation*. Because practical implementations can bridge these stages, we classify works based on the primary task component they target; systems that independently evaluate both stages or formalize a complete pipeline are assigned multiple relevant categories.

### A. Hypothesis Selection

These tasks instantiate Stage II. The model is given candidate hypotheses—either as an explicit set of alternatives or as individual hypotheses paired with evidence—and must judge which explanation is most plausible or whether a particular explanation is acceptable.

- **A1. Hypothesis Scoring and Selection.** The model evaluates multiple candidate explanations and either selects the best one or ranks them by plausibility. This is the most common benchmark formulation of abductive reasoning, especially in multiple-choice settings. Canonical examples include ART/$\alpha$NLI (Bhagavatula et al., 2020), as well as domain-specific variants such as differential-diagnosis prediction in DDXPlus, where models prioritize candidate diagnoses given patient findings (Tchango et al., 2022).

- **A2. Single-Hypothesis Evaluation.** The model evaluates one proposed hypothesis against the available evidence and determines whether it is a plausible explanation. The output may be a binary judgment, a label, or a graded plausibility assessment. This formulation appears in tasks such as CauseJudger (He & Lu, 2024), where the model judges whether a proposed cause is consistent with given premises, and in defeasible inference settings such as $\delta$-NLI (Rudinger et al., 2020), where the model assesses how new evidence changes the status of a candidate conclusion.

### B. Hypothesis Generation

These tasks instantiate Stage I. Rather than choosing among predefined options, the model must produce a new hypothesis—for example, an event, fact, mechanism, or rule—that would make the observations intelligible.

- **B1. Explanation Generation.** The model generates a free-form explanation for the observed situation, often as narrative completion, event infilling, or causal elaboration. These tasks are typically open-ended and do not require the model to complete an explicit formal theory. Representative examples include $\alpha$NLG on ART (Bhagavatula et al., 2020) and UNcommonsense (Zhao et al., 2024).

- **B2. Knowledge Completion (Facts, Rules, Programs).** The model generates missing knowledge that, once added to a context, rule base, or world model, makes the observation derivable or coherent. Compared with B1, these tasks are usually constrained by a more explicit structure, such as a rule system, proof context, or knowledge graph. Examples include the abductive setting in ProofWriter (Tafjord et al., 2021), AbductionRules (Young et al., 2022), and knowledge-graph abduction tasks that generate missing logical paths or relations (Bai et al., 2024; Gao et al., 2026b).

**Full-pipeline frameworks (Stage I + Stage II).** Some work targets the full abductive pipeline by combining hypothesis generation with an explicit downstream evaluation step. In some cases, these systems generate multiple candidate hypotheses and compare them using explanatory virtues such as consistency, diversity, generalizability, or simplicity (He et al., 2025a; Sun & Saparov, 2025). In other cases, hypothesis generation is followed by task-specific validation, ranking, or decision-oriented assessment within a broader application pipeline (Shi et al., 2023; Hong et al., 2024). We therefore treat such work not as a separate task

category, but as an integrative formulation built by combining the core task types above: the generation step typically falls under B1 or B2, while the evaluation step typically falls under A1 or A2.

**Interactive multi-step settings.** In this survey, we define abduction at the level of an individual explanatory step: given a set of observations, the system proposes one or more candidate hypotheses and/or evaluates how well candidate hypotheses account for those observations. This step-level view keeps the taxonomy clear, while still applying naturally to broader interactive multi-step settings (Shi et al., 2023; Hong et al., 2024). In such settings, a system may contain several abductive steps, and those steps may fall into different categories in our taxonomy; other steps may instead involve retrieval, deduction, verification, or action selection. In this sense, multi-step or action-oriented benchmarks are not outside the scope of the taxonomy, but are better understood as higher-level frameworks composed of one or more abductive task instances.

**Modality: textual vs. multimodal abduction.** The same task formulations also appear in multimodal settings, where observations may include images, videos, or other perceptual inputs. Examples include visual abductive reasoning over images or video (Hessel et al., 2022; Liang et al., 2022) and multimodal action-chain abduction (Li et al., 2023). These settings add grounding challenges, but they still fit the same underlying distinction between hypothesis generation and hypothesis selection.

## 3.2 By Dataset Type

Along the dataset axis, abductive reasoning benchmarks differ mainly in the kind of background knowledge they assume. We keep a broad split between (A) commonsense datasets, which rely primarily on implicit everyday world knowledge, and (B) expert or formal datasets, which depend on specialized knowledge or an explicitly specified structure. As a practical rule, a dataset belongs to Family B when solving it requires domain expertise or reasoning within a formal system, even if the inputs themselves are written in natural language. While datasets may incorporate overlapping elements, we categorize them based on their dominant knowledge requirement, assigning multiple categories only when a work introduces distinct subsets spanning across different dataset families.

### A. Commonsense Abduction Datasets

These datasets center on everyday physical, social, or causal situations. The needed background knowledge is usually implicit rather than formally provided, and the inputs are typically short narratives, scenes, or multimodal observations.

- **A1. Text Commonsense.** These datasets present natural-language situations from ordinary life and ask for an explanation of an event, state, or transition. The missing hypothesis is usually a cause, intermediate event, motivation, or enabling condition that makes the observations cohere. Canonical examples include ART/$\alpha$NLI, which frames abduction as selecting the most plausible hypothesis connecting two observations (Bhagavatula et al., 2020); e-CARE, which emphasizes causal explanation in everyday scenarios (Du et al., 2022); and UNcommonsense, which focuses on surprising or low-prior outcomes that require less stereotypical explanations (Zhao et al., 2024). Long-form mystery settings such as True Detective extend the same commonsense abductive pattern to longer narratives with dispersed clues (Del & Fishel, 2023).

- **A2. Multimodal Commonsense.** Here the underlying knowledge is still everyday commonsense, but the observations include images or videos in addition to text. The abductive step is to infer a plausible hidden event, cause, or action chain that explains what is partially observed. Representative examples include Sherlock for image-based visual abduction (Hessel et al., 2022), Visual Abductive Reasoning for inferring missing events from incomplete video observations (Liang et al., 2022), VideoABC for real-world video abduction (Zhao et al., 2022), and MAR for multimodal action-chain abduction (Li et al., 2023).

### B. Expert & Formal Abduction Datasets

These datasets move beyond everyday world knowledge. Some are grounded in specialized domains such as medicine or law; others are defined by explicit logical, programmatic, or graph-structured systems. In both cases, the explanation must respect stronger domain or formal constraints than in commonsense settings.

- **B1. Applied Domain (Science, Diagnosis & Law).** In this family, observations come from an expert domain and plausible explanations must conform to domain-specific constraints. Med-CaseReasoning is a clear example: models are given clinical case presentations and must infer diagnoses in a medically grounded setting (Wu et al., 2025b). Legal abduction benchmarks such as L'ART similarly places explanatory reasoning in a domain with more explicit normative and logical constraints than ordinary commonsense narrative tasks (Nguyen et al., 2023).

- **B2. Logic & Formal Reasoning.** These datasets are defined by an explicit formal theory, rule base, or constrained reasoning system. The abductive task is typically to recover a missing fact, premise, or hypothesis that makes a target conclusion derivable or coherent under the given rules. Representative examples include AbductionRules (Young et al., 2022), the abductive setting in ProofWriter (Tafjord et al., 2021), CauseJudger (He & Lu, 2024), and UniADILR (Sheng et al., 2025). What unifies them is not a shared task format, but the presence of an explicit formal or logical structure that bounds the hypothesis space.

- **B3. Programs & Rule Learning.** In this family, the latent explanatory structure is best viewed as a program or rule system. Rather than explaining an observation in free text, the model must infer missing program-like structure consistent with the observed behavior. Mini-ARC (Kim et al., 2022), a 5x5 compact version of the ARC (Chollet, 2019), is the clearest benchmark example, since each task requires inferring a transformation rule from input–output grids.

- **B4. Knowledge Graph Abduction.** These datasets embed abduction in graph-structured knowledge. Observations are entities, triples, or subgraphs, and the goal is to propose symbolic hypotheses—such as missing relations, logical paths, or explanatory rule patterns—that best account for what is observed. Representative examples include formal abductive hypothesis generation over knowledge graphs (Bai et al., 2024) and KG validation settings that use abductive evidence to support or challenge candidate triples (Du et al., 2019). Compared with B2, the key distinction is that the background structure is relational and graph-based rather than a standalone rule system.

### 3.3 By Methodology

Methodological differences in this literature are best understood in terms of how abductive behavior is obtained from the model: by steering inference-time behavior, adapting model parameters, adding external knowledge, distributing the reasoning process across interacting agents, or coupling neural models with symbolic machinery. We therefore retain five broad methodological families (M1–M5). These categories are not mutually exclusive; while complex systems frequently combine several techniques, we classify works under the dominant methodology that primarily drives their abductive reasoning. Papers that introduce and evaluate multiple independent frameworks are explicitly assigned all applicable categories.

### M1. Prompt Engineering Approaches

Prompt engineering elicits abductive behavior at inference time without changing model parameters. In this literature, it appears mainly as decomposition prompts that separate observation reading, candidate-hypothesis generation, and hypothesis comparison, or as criteria-guided prompts that ask the model to judge explanations in terms such as consistency, parsimony, or plausibility (Liu et al., 2024; He & Lu, 2024; Dalal et al., 2024). Prompting is attractive because it is lightweight and easy to transfer across models, and it is often the first method used to probe abductive ability in new settings. Its role, however, is primarily to *steer* existing model behavior rather than to supply new knowledge or task-specific reasoning competence.

## M2. Model Training

Training-based methods adapt model parameters to abductive tasks through supervised fine-tuning (SFT) or related task-specific objectives. This remains the dominant paradigm in benchmark-driven research, including commonsense explanation tasks such as ART and UNcommonsense, as well as structured settings like AbductionRules, DDXPlus, and controllable knowledge-graph hypothesis generation (Bhagavatula et al., 2020; Zhao et al., 2024; Young et al., 2022; Tchango et al., 2022; Gao et al., 2026b). Most prior work relies on SFT over annotated premise–observation–hypothesis examples, which typically yields strong in-distribution performance.

A smaller line of work explores alternative optimization strategies. For example, RLF-KG applies reinforcement learning with PPO, rewarding hypotheses based on how well their execution reconstructs observed facts (Bai et al., 2024). GEAR instead uses Direct Preference Optimization (DPO), evaluating generated hypotheses through criteria such as consistency, generalizability, and diversity without relying on reference answers (He et al., 2025a).

## M3. Knowledge-Augmented Approaches

Knowledge-augmented methods support abduction with information that is not left entirely to model parameters. In the current literature, this usually takes the form of retrieval over documents or the integration of structured resources such as knowledge graphs, so that generated or selected hypotheses remain better grounded in domain constraints (Gao et al., 2025; Lin, 2025). This family is especially natural in expert settings, where plausible explanations depend on specialized background knowledge that should be retrieved or explicitly represented rather than implicitly memorized. Its main benefit is improved grounding, but performance is correspondingly sensitive to the quality of retrieval, graph construction, and evidence selection.

## M4. Multi-Agent Frameworks

Multi-agent frameworks distribute abductive reasoning across multiple interacting components rather than asking a single model call to do everything at once. Typical systems assign distinct roles to generators, critics, retrievers, or synthesizers, and let these agents interact sequentially or iteratively while refining candidate explanations (He et al., 2023; Hong et al., 2024). This design is most useful when the abductive process itself is treated as multi-stage—for example, when hypothesis proposal, evidence gathering, and hypothesis evaluation are made explicit. While individual components within these frameworks often rely on prompt-based instructions (M1), M4 is distinguished by the explicit distribution of tasks and iterative communication protocols among distinct agent roles, whereas M1 focuses on steering a single model's execution path. Compared with single-agent prompting, these systems offer greater modularity and clearer process structure, but they also introduce extra orchestration cost and more opportunities for pipeline error.

## M5. Hybrid Neuro-Symbolic Approaches

Hybrid neuro-symbolic approaches combine neural language modeling with symbolic or formally structured reasoning components. For example, pipelines that translate natural-language hypotheses into symbolic forms for verification or constrained backward reasoning (Li et al., 2026; Hong et al., 2024). The central idea is not merely to generate explanations fluently, but to represent them in a form that can be constrained, verified, or learned against an explicit formal structure. These methods are particularly relevant when the hypothesis space is rule-like, program-like, or otherwise machine-checkable, though they remain less common than prompting- or fine-tuning-based approaches in current LLM-centered abduction work.

### 3.4 By Evaluation Approach

Abductive reasoning work also differs in the signal used to assess model performance. We group evaluation approaches into three broad families: (A) *accuracy-based evaluation*, where the final outcome is a discrete correct/incorrect judgment; (B) *non-accuracy-based evaluation*, where outputs are scored automatically but not reduced to a single correctness label; and (C) *human evaluation*, where people directly judge the quality of explanations. Because many papers report a suite of metrics, we classify them based on the primary evalu-

ation signal used to establish the system's efficacy, assigning multiple categories when a study fundamentally relies on distinct, independent evaluation paradigms.

## A. Accuracy-based Evaluation

Accuracy-based evaluation uses a discrete success signal. This is most common when abduction is formulated as selecting among candidate hypotheses, but it also appears in structured settings where a proposed explanation can be checked against formal constraints and then marked correct or incorrect.

- **A1. Direct-match Evaluation.** Direct-match evaluation scores the model output against an expected answer using metrics such as accuracy, exact match, or F1. It is the standard choice for hypothesis-selection benchmarks with a closed answer space. Canonical examples include $\alpha$NLI in ART, where the model chooses the more plausible hypothesis connecting two observations (Bhagavatula et al., 2020); DDXPlus, where systems rank or predict diagnoses from patient findings (Tchango et al., 2022); and long-context selection settings such as True Detective (Del & Fishel, 2023). The common feature is that evaluation is decided by whether the system arrives at the designated answer, rather than by separately scoring the quality of an open-ended explanation.

  **A2. Verification-based Evaluation.** Verification-based evaluation also produces a discrete success signal, but correctness is established by checking whether a generated or completed hypothesis satisfies external constraints. In formal abduction, this may mean that the proposed premise, rule, or explanation makes a target conclusion derivable, remains logically consistent, or is validated by a symbolic solver. This pattern appears most clearly in structured reasoning settings such as ProofWriter-style abduction and logic-based abduction frameworks, where hypotheses are judged by whether they support a valid derivation under an explicit rule system (Tafjord et al., 2021). The distinguishing property of this category is that the final score still reduces to correctness, even though it is obtained through verification rather than direct answer matching.

## B. Non-accuracy-based Evaluation

Non-accuracy-based evaluation remains automatic, but it does not assume that abductive quality is exhausted by a single correct answer. This is especially common for open-ended generation, where several explanations may be reasonable and evaluation instead focuses on similarity to references or on intrinsic properties of the generated hypotheses.

- **B1. Reference-based Similarity Evaluation.** Reference-based similarity evaluation compares a generated explanation against one or more gold references using lexical, semantic, or structural similarity measures. In ART's $\alpha$NLG setting, generated hypotheses are evaluated against human-written explanations with metrics such as BLEU, ROUGE, CIDEr, and BERTScore (Bhagavatula et al., 2020). The same general pattern appears in later open-ended commonsense generation work such as UNcommonsense (Zhao et al., 2024), as well as in more structured generation benchmarks such as AbductionRules, where the target is a missing fact or rule completion rather than a free-form narrative (Young et al., 2022). The score is automatic and reference-grounded, but it is not interpreted as exact correctness.

- **B2. Reference-free Quality Evaluation.** Reference-free quality evaluation scores hypotheses without relying on a single gold explanation. Instead, it assesses generated outputs using abductive criteria such as plausibility, consistency, parsimony, diversity, or predictive usefulness. GEAR is a clear recent example: it evaluates sets of generated hypotheses through consistency, generalizability, and diversity, explicitly avoiding dependence on reference answers (He et al., 2025a). IBE-Eval follows a similar direction by scoring explanations with feature-based criteria motivated by inference to the best explanation, such as coherence, consistency, and simplicity (Dalal et al., 2024). This category is useful when a correct explanation can be expressed in multiple valid ways and reference matching would understate the range of acceptable abductive outputs.

## C. Human Evaluation

Human evaluation uses people as the evaluation signal. Annotators may rate explanations, rank alternatives, make pairwise preference judgments, or provide qualitative error analyses. This is particularly important in open-ended abduction, where automatic metrics often miss subtle differences in plausibility, informativeness, or explanatory adequacy.

In practice, human evaluation is most often used to assess generated explanations rather than closed-form selections. ART reports human judgments for abductive explanation generation in addition to automatic metrics (Bhagavatula et al., 2020), and UNcommonsense likewise compares model-generated explanations with human-written ones using direct human judgments (Zhao et al., 2024). Human assessment also serves as a useful anchor for newer automatic evaluators: for example, IBE-Eval reports agreement with human preferences to show that its reference-free scores track perceived explanatory quality (Dalal et al., 2024). We treat such cases as human evaluation only when humans directly judge model outputs; using human judgments merely to validate an automatic metric does not by itself change the primary evaluation category of a benchmark.

Figure 3 provides a visual overview of our proposed taxonomy for categorizing the literature on abductive reasoning in LLMs.

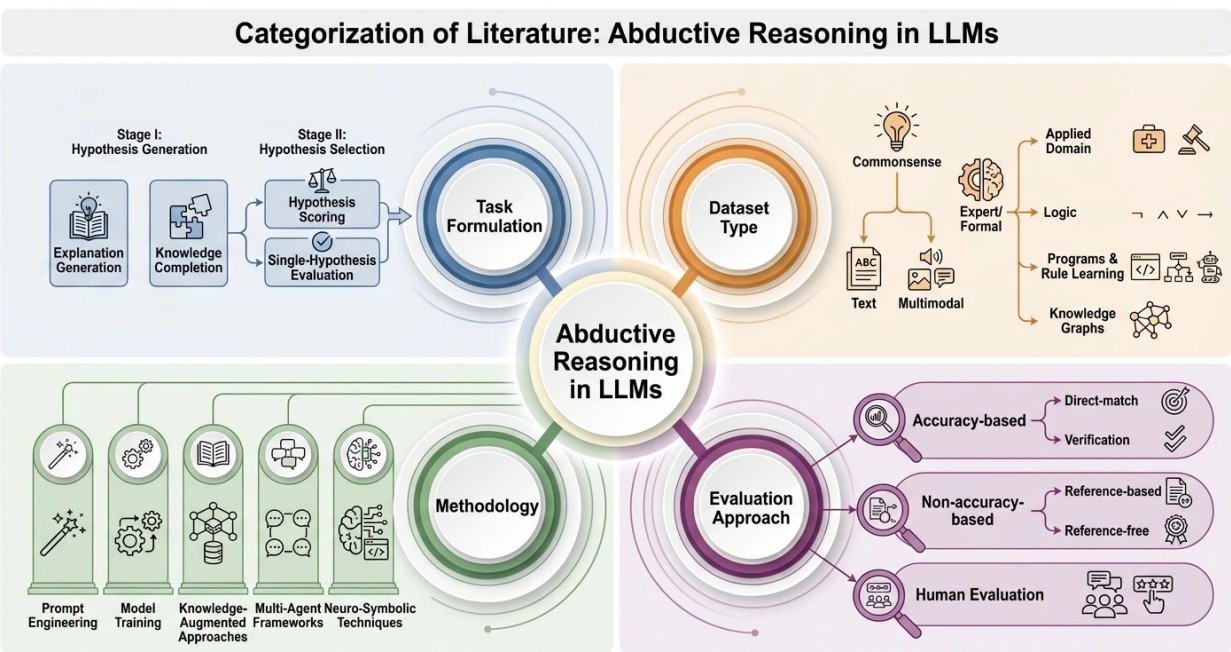

Figure 3: Categorization of literature on abductive reasoning in LLMs. The taxonomy is divided into four main dimensions: Task Formulation, Dataset Type, Methodology, and Evaluation Approach.

### 3.5 Categorization of Works

Section 3 has organized the literature along four complementary dimensions: task formulation, dataset type, methodology, and evaluation approach. Table 2 brings these dimensions together to provide a compact view of prior work. (This categorization includes papers analyzed up to the end of February 2026.)

In the task dimension, we distinguish between hypothesis selection (A), where models assess or choose among candidate explanations, and hypothesis generation (B), where models produce explanatory hypotheses. On the data side, we separate commonsense datasets (A) from expert and formal datasets (B), including applied-domain settings such as science, diagnosis, and law, along with more explicitly structured settings such as

logic and formal reasoning, programs and rule learning, and knowledge graph abduction. Methodologically, we group studies into prompt engineering (M1), model training (M2), knowledge-augmented approaches (M3), multi-agent frameworks (M4), and hybrid neuro-symbolic approaches (M5). Evaluation is organized into (A) accuracy-based evaluation, (B) non-accuracy-based evaluation, and (C) human evaluation.

Because many papers address only part of the abductive process, the table also indicates whether a work focuses on generation, selection, or both. We use B+A to denote a genuine full pipeline, in which hypotheses are first generated and then evaluated or ranked. By contrast, when A and B appear separately in the task column, this simply indicates that both task types are present in the paper, not necessarily that they are integrated into a single end-to-end pipeline.

Finally, some studies combine abductive reasoning with other forms of inference, such as deductive or inductive components. In such cases, the table classifies only the abductive component so that the comparison remains aligned with the scope of this survey.

Table 2: Unified categorization of abductive reasoning works.

| Paper | Task | Dataset | Method | Evaluation |
|---|---|---|---|---|
| Abductive Commonsense Reasoning (ART) (Bhagavatula et al., 2020) | A1, B1 | A1 | M2 | A1, B1, C |
| UniADILR (Sheng et al., 2025) | A1 | B2 | M1, M2 | A1 |
| CauseJudger (He & Lu, 2024) | A2 | B2 | M1 | A1 |
| Thinking Like a Skeptic: Defeasible Inference in Natural Language (Rudinger et al., 2020) | A2 | A1 | M2 | A1 |
| DDXPlus (Tchango et al., 2022) | A1 | B1 | M2 | A1 |
| An Incomplete Loop: Instruction Inference, Instruction Following, and In-context Learning in Language Models (Liu et al., 2024) | B2 + A1 (full pipeline) | B3 | M1 | A1 |
| e-CARE (Du et al., 2022) | A1, B1 | A1 | M2 | A1, B1, C |
| Self-Consistent Narrative Prompts on Abductive Natural Language Inference (Chan et al., 2023) | A1 | A1 | M1 | A1 |
| UNcommonsense Reasoning (Zhao et al., 2024) | B1 | A1 | M2 | B1, C |
| ProofWriter (Tafjord et al., 2021) | B2 | B2 | M2 | A2 |
| RuleTaker (Clark et al., 2020) | A2 | B2 | M2 | A1 |
| True Detective (Del & Fishel, 2023) | A1 | A1 | M1 | A1 |
| Moral Stories (Emelin et al., 2021) | B1 | A1 | M2 | B1, C |
| Language Models Do Not Follow Occam's Razor: A Benchmark for Inductive and Abductive Reasoning (Sun & Saparov, 2025) | B2 | B2 | M1 | A2, B2 |
| SocialIQA (Sap et al., 2019) | A1 | A1 | M2 | A1 |
| L2R²: Leveraging Ranking for Abductive Reasoning (Zhu et al., 2020) | A1 | A1 | M2 | A1 |
| Social Commonsense Reasoning with Multi-Head Knowledge Attention (Paul & Frank, 2020) | A1 | A1 | M2, M3 | A1 |

| Paper | Task | Data | Method | Eval |
|---|---|---|---|---|
| Are Large Language Models Really Good Logical Reasoners? A Comprehensive Evaluation and Beyond (Xu et al., 2025a) | A1, A2, B1, B2 | A1, B2 | M1 | A1, B1, B2, C |
| Evaluating the Logical Reasoning Abilities of Large Reasoning Models (Liu et al., 2025) | A1 | A1 | M1 | A1 |
| How well do SOTA legal reasoning models support abductive reasoning? (Nguyen et al., 2023) | A2 | B1 | M1 | A1 |
| AbductionRules: Training Transformers to Explain Unexpected Inputs (Young et al., 2022) | B2 | B2 | M2 | B1 |
| Language Models Can Improve Event Prediction by Few-Shot Abductive Reasoning (Shi et al., 2023) | B1 + A1 (full pipeline) | B1 | M1, M3 | A1 |
| Visual Abductive Reasoning (Liang et al., 2022) | B1 | A2 | M2 | B1 |
| The Abduction of Sherlock Holmes: A Dataset for Visual Abductive Reasoning (Hessel et al., 2022) | A1, A2 | A2 | M2 | A1 |
| LogiDynamics: Unraveling the Dynamics of Inductive, Abductive and Deductive Logical Inferences in LLM Reasoning (Zheng et al., 2025) | B2 | A1, A2, B3 | M1 | A1 |
| LEGO: A Multi-agent Collaborative Framework with Role-playing and Iterative Feedback for Causality Explanation Generation (He et al., 2023) | B1 | A1 | M4 | B1, C |
| ArgMed-Agents: Explainable Clinical Decision Reasoning with Large Language Models via Argumentation Schemes (Hong et al., 2024) | B1 + A1 (full pipeline) | B1 | M4, M5 | A1, C |
| MedCaseReasoning: Evaluating and learning diagnostic reasoning from clinical case reports (Wu et al., 2025b) | B1 | B1 | M1, M2 | A1, B1 |
| ER-REASON: A Benchmark Dataset for LLM-Based Clinical Reasoning in the Emergency Room (Mehandru et al., 2025) | B1 | B1 | M1 | B1, C |
| GEAR: A General Evaluation Framework for Abductive Reasoning (He et al., 2025a) | B2 | B3 | M1, M2 | A2, B2 |
| Leveraging Medical Knowledge Graphs Into Large Language Models for Diagnosis Prediction: Design and Application Study (Gao et al., 2025) | A1, B1 | B1 | M3 | A1, B1 |
| VideoABC: A Real-World Video Dataset for Abductive Visual Reasoning (Zhao et al., 2022) | A1 | A2 | M2 | A1 |
| Multi-modal Action Chain Abductive Reasoning (MAR) (Li et al., 2023) | B1 | A2 | M2 | B1 |

| Paper | Task | Data | Method | Eval |
|---|---|---|---|---|
| Active Reasoning in an Open-World Environment (Xu et al., 2023) | A1 | A2 | M3, M4 | A1 |
| Black Swan: Abductive and Defeasible Video Reasoning in Unpredictable Events (Chinchure et al., 2025) | A1, A2, B1 | A2 | M1 | A1, B1 |
| Validation of Growing Knowledge Graphs by Abductive Text Evidences (Du et al., 2019) | A1, B2 | B4 | M3, M2 | A1 |
| ExplanationLP (Thayaparan et al., 2020) | A1 | B1 | M2, M3, M5 | A1 |
| Inference to the Best Explanation in Large Language Models (IBE-Eval) (Dalal et al., 2024) | A1 | A1 | M1 | B2, C |
| Towards LogiGLUE: A Brief Survey and A Benchmark for Analyzing Logical Reasoning Capabilities of Language Models (Luo et al., 2023) | A1, B2 | A1, B2 | M2 | A1, A2 |
| IDEA: Enhancing the Rule Learning Ability of Large Language Model Agent through Induction, Deduction, and Abduction (He et al., 2025b) | B2 | B3 | M4 | A1 |
| From Hypothesis to Premises: LLM-based Backward Logical Reasoning with Selective Symbolic Translation (Li et al., 2026) | A2 | B2 | M5 | A1, A2 |
| Large Language Models as Biomedical Hypothesis Generators: A Comprehensive Evaluation (Qi et al., 2024) | B1 | B1 | M1, M2, M3, M4 | B2, C |
| Abductive Commonsense Reasoning Exploiting Mutually Exclusive Explanations (Zhao et al., 2023) | A1 | A1 | M2 | A1 |
| Back to the Future: Unsupervised Backprop-based Decoding for Counterfactual and Abductive Commonsense Reasoning (Qin et al., 2020) | B1 | A1 | M1 | B1, C |
| Dynamic knowledge correction via abductive for domain question answering (Zhou et al., 2026) | B2 | B1 | M3 | B1 |
| WHODUNIT (Gupta, 2025) | A1 | A1 | M1 | A1 |
| DetectiveQA (Xu et al., 2025b) | A1 | A1 | M1 | A1, B1 |
| Reasoning Like a Doctor: Improving Medical Dialogue Systems via Diagnostic Reasoning Process Alignment (Xu et al., 2024) | B1 | B1 | M2, M3 | A1, C |
| Assessing the Reasoning Capabilities of LLMs in the context of Evidence-based Claim Verification (Dougrez-Lewis et al., 2025) | A2 | A1 | M1 | A1, B1 |
| Disentangling Logic: The Role of Context in Large Language Model Reasoning Capabilities (Hua et al., 2025) | B2 | B2 | M2 | A1, A2 |
| PARADISE (Uzunoğlu et al., 2024) | A1 | A1 | M1, M2 | A1 |

| Paper | Task | Data | Method | Eval |
|---|---|---|---|---|
| The Magic of IF: Investigating Causal Reasoning Abilities in Large Language Models of Code (Liu et al., 2023) | B1 | A1 | M1 | B1, C |
| Abductive Inference in Retrieval-Augmented Language Models: Generating and Validating Missing Premises (Lin, 2025) | B2 + A1 (full pipeline) | B2 | M3 | A1, C |
| DixitWorld (Mo et al., 2025) | B1 + A1 (full pipeline) | A2 | M4 | A1 |
| Leveraging Symbolic Knowledge Bases for Commonsense Natural Language Inference Using Pattern Theory (Aakur & Sarkar, 2023) | A2 | A1 | M5 | A1 |
| Epistemology of Language Models: Do Language Models Have Holistic Knowledge? (Kim & Thorne, 2024) | B1 | B1 | M1 | B1, C |
| Doing Experiments and Revising Rules with Natural Language and Probabilistic Reasoning (Piriyakulkij et al., 2024) | B2 + A1 (full pipeline) | B3 | M5 | A1 |
| Unifying Deductive and Abductive Reasoning in Knowledge Graphs with Masked Diffusion Model (DARK) (Gao et al., 2026a) | B2 + A1 (full pipeline) | B4 | M5 | A1, A2 |
| Graph-PReFLexOR (Buehler, 2025) | B2 | B1 | M5 | B2 |
| AbductiveMLLM: Boosting Visual Abductive Reasoning Within MLLMs (Chang et al., 2026) | B1 + A1 (full pipeline) | A2 | M2 | B1 |
| Controllable Logical Hypothesis Generation for Abductive Reasoning in Knowledge Graphs (Gao et al., 2026b) | B2 | B4 | M2 | B1, B2 |
| From We to Me: Theory Informed Narrative Shift with Abductive Reasoning (Patil et al., 2026) | B2 + A1 (full pipeline) | A1 | M5 | B1, B2 |

## 4 Experiments and Analysis

### 4.1 Benchmarking

To complement the taxonomy developed in Section 3, we include a compact empirical study that instantiates its main distinctions in practice. Our goal is not to establish a universal leaderboard for abductive reasoning, but to ground the survey in a benchmark suite that reflects the scope of our working definition (Section 2.2.1), spans both broad dataset families introduced in Section 3.2, and covers several evaluation styles. Appendix 9.1 provides the benchmark formulations, prompt schemas, and metric definitions used in this study.

The suite is organized to cover both stages of abductive reasoning. For Stage II selection, we include short commonsense bridging tasks (ART and e-CARE (Bhagavatula et al., 2020; Du et al., 2022)), diagnosis ranking under medical constraints (DDXPlus (Tchango et al., 2022)), and longer narrative inference with dispersed clues (True Detective (Del & Fishel, 2023) and the murder subset of MuSR (Sprague et al., 2024)). For Stage I generation, we include open-text explanation tasks (ART, e-CARE, and UNcommonsense (Bhagavatula et al., 2020; Du et al., 2022; Zhao et al., 2024)), open-ended diagnosis generation (DDXPlus), and formally constrained missing-premise completion (ProofWriter, AbductionRules, and NeuLR (Tafjord et al., 2021; Young et al., 2022; Xu et al., 2025a)).

We evaluate a mix of contemporary open-weight and proprietary LLMs, with open-weight models ranging from 3B to 72B parameters; the exact model list appears in Tables 3–6. Each benchmark is run with a single task-specific direct instruction template held fixed across models. Closed-form tasks are scored with accuracy or top-$k$ ranking metrics. Open-ended generation is evaluated using a combination of task-validity checks, reference-based similarity, judge-based pairwise comparison, and exact-match or F1-style metrics when the target admits structural verification. Because the experiments use the splits and subsets specified in Appendix 9.1, comparisons with externally reported results should take these evaluation choices into account. In particular, some tasks are evaluated on fixed subsets rather than full test sets, and several open-ended metrics rely on LLM-as-a-judge evaluation using Gemini 3 Flash Preview. These factors can affect both statistical uncertainty and comparability across studies, so cross-study comparisons should be made with appropriate caution.

| Model | ART | e-CARE | DDXPlus | | True Detective | MuSR |
|---|---|---|---|---|---|---|
| | | | Top-1 | Hit@3 | | |
| Qwen2.5 3B | 65.8 | 78.6 | 41.7 | 70.0 | 21.9 | 54.8 |
| Qwen2.5 7B | 75.8 | 83.0 | 54.5 | 86.2 | 26.1 | 58.0 |
| Qwen2.5 72B | 83.6 | 87.0 | 70.5 | 93.2 | 27.2 | 62.8 |
| Qwen3 8B | 73.0 | 83.8 | 60.5 | 89.2 | 34.5 | 54.8 |
| Qwen3 32B | 78.2 | 86.2 | 73.7 | 94.7 | 40.8 | 58.4 |
| Llama3.1 8B | 64.2 | 76.0 | 55.7 | 86.0 | 28.2 | 52.4 |
| Llama3.1 70B | 83.6 | 87.2 | 71.0 | 93.7 | 29.8 | 62.0 |
| Llama3.3 70B | 85.2 | 86.2 | 70.2 | 95.2 | 31.4 | 63.2 |
| DeepSeek-V3.2 | 82.0 | 84.2 | 73.2 | 95.7 | 34.5 | 61.2 |
| GPT-4o | **87.2** | 87.4 | 75.2 | 96.7 | 39.7 | **68.0** |
| GPT-5.4 | **87.2** | **88.0** | **79.75** | **98.7** | **42.9** | 60.8 |
| *Human avg* | *91.4* | *92.0* | *N/A* | *N/A* | *47.0* | *92.1* |

Table 3: Stage II (selection) results on the selected abductive benchmarks. DDXPlus is evaluated as a diagnosis-ranking task, so we report Top-1 and Hit@3 rather than simple accuracy. Reported human performance is unavailable for DDXPlus. For True Detective, the average human solve rate is 47%, although top human solvers exceed 80%.

| Model | ART | e-CARE | DDXPlus | UNcommonsense | ProofWriter | AbductionRules | NeuLR |
|---|---|---|---|---|---|---|---|
| | Val. | Val. | Hit@3 | WR-C+L | Acc. | Acc. | Acc. |
| Qwen2.5 3B | 63.2 | 80.2 | 32.5 | 6.5 | 0.0 | 13.7 | 10.0 |
| Qwen2.5 7B | 71.5 | 86.5 | 41.7 | 16.0 | 1.2 | 21.0 | 15.2 |
| Qwen2.5 72B | 90.0 | 94.0 | 51.0 | 21.0 | 9.7 | 45.2 | 43.2 |
| Qwen3 8B | 74.2 | 85.0 | 46.5 | 13.0 | 0.0 | 35.0 | 16.8 |
| Qwen3 32B | 83.5 | **95.0** | 48.7 | 24.0 | 3.5 | 46.2 | 31.2 |
| Llama3.1 8B | 74.5 | 85.0 | 41.0 | 14.0 | 1.5 | 25.7 | 10.8 |
| Llama3.1 70B | 92.7 | 92.7 | 49.5 | 37.5 | 7.5 | 46.0 | 42.8 |
| Llama3.3 70B | 91.7 | 90.7 | 49.2 | 41.0 | 4.7 | 47.7 | 38.0 |
| DeepSeek-V3.2 | 92.2 | 90.0 | 55.7 | 32.5 | 5.2 | 71.7 | 45.6 |
| GPT-4o | 92.5 | 94.5 | 57.5 | 33.0 | 14.0 | 50.7 | 52.4 |
| GPT-5.4 | **94.7** | 92.2 | **63.7** | **61.5** | **21.5** | **84.5** | **70.8** |

Table 4: Stage I (generation) results on the primary task metrics. **Val.** denotes format/task validity for ART and e-CARE. **Hit@3** on DDXPlus is top-3 hit rate after strict judge-based disease matching. **WR-C+L** on UNcommonsense is pairwise win rate against enhanced crowd-written references. **Acc.** denotes task accuracy under the benchmark's standard evaluation protocol.

| Model | ART WR-H | e-CARE CEQ. | DDXPlus Set-F1@3 | UNcommonsense WR-C | ProofWriter F1 | AbductionRules Tok-F1 | NeuLR CLS. |
|---|---|---|---|---|---|---|---|
| Qwen2.5 3B | 36.5 | 0.045 | 16.6 | 16.5 | 0.3 | 68.9 | 61.7 |
| Qwen2.5 7B | 44.2 | 0.056 | 18.4 | 29.0 | 5.3 | 74.1 | 66.5 |
| Qwen2.5 72B | 68.7 | 0.051 | 22.2 | 47.5 | 26.5 | 89.2 | 79.9 |
| Qwen3 8B | 48.0 | 0.052 | 22.3 | 26.0 | 3.2 | 75.5 | 60.7 |
| Qwen3 32B | 62.5 | 0.056 | 22.5 | 43.5 | 30.5 | 88.7 | 73.7 |
| Llama3.1 8B | 45.2 | 0.038 | 19.2 | 31.0 | 17.8 | 74.2 | 61.5 |
| Llama3.1 70B | **73.7** | 0.061 | 22.4 | 55.5 | 39.8 | 88.5 | 77.4 |
| Llama3.3 70B | 72.2 | 0.057 | 22.8 | 56.5 | 35.9 | 88.9 | 73.4 |
| DeepSeek-V3.2 | 67.7 | 0.060 | 24.5 | 54.5 | 34.1 | 94.1 | 78.6 |
| GPT-4o | 68.7 | 0.058 | 25.7 | 56.0 | 34.5 | 91.8 | 81.3 |
| GPT-5.4 | 72.7 | **0.071** | **27.9** | **84.0** | **61.6** | **96.4** | **89.3** |

Table 5: Complementary quality metrics for Stage I generation. **WR-H** is ART win rate against the human reference. **CEQ.** is causal explanation quality on e-CARE. **Set-F1@3** on DDXPlus measures set overlap for the top-3 predicted diseases after strict judge-based matching. **WR-C** is win rate against crowd-written references on UNcommonsense. **F1** for ProofWriter is proof-generation F1, **Tok-F1** is token-level F1 on AbductionRules, and **CLS.** is mean character-level Levenshtein similarity on NeuLR.

| Model | ART | | | e-CARE | | UNcommonsense |
|---|---|---|---|---|---|---|
| | **BLEU-4** | **ROUGE-L** | **BERTScore** | **BLEU** | **ROUGE-L** | **BERTScore** |
| Qwen2.5 3B | 2.99 | 18.1 | 53.2 | 19.7 | 9.6 | 89.0 |
| Qwen2.5 7B | 2.33 | 21.6 | 55.1 | 26.4 | 15.9 | 89.2 |
| Qwen2.5 72B | 3.93 | 23.1 | 56.4 | 26.7 | 18.0 | 89.5 |
| Qwen3 8B | 2.51 | 19.5 | 54.0 | 26.2 | 16.5 | 89.5 |
| Qwen3 32B | 2.43 | 20.4 | 54.5 | 27.2 | 18.7 | 89.6 |
| Llama3.1 8B | 3.11 | 20.8 | 54.4 | 22.6 | 14.8 | 88.9 |
| Llama3.1 70B | 3.79 | 20.5 | 54.4 | 28.7 | 21.7 | 89.1 |
| Llama3.3 70B | 3.66 | 20.6 | 54.3 | 30.7 | 22.1 | 87.9 |
| DeepSeek-V3.2 | 3.86 | 21.9 | 55.2 | 26.1 | 17.1 | **89.9** |
| GPT-4o | 3.76 | **23.4** | **57.0** | 27.6 | 19.7 | 89.5 |
| GPT-5.4 | **4.22** | 20.4 | 55.5 | **32.4** | **23.3** | 88.4 |

Table 6: Reference-based similarity scores for the open-ended generation tasks where such metrics are informative. For ART we report BLEU-4, ROUGE-L, and BERTScore; for e-CARE, BLEU and ROUGE-L; for UNcommonsense, BERTScore. We report these separately because lexical or embedding similarity does not always track judged explanatory quality.

## Main Empirical Picture

Across the suite, short closed-form selection tasks appear substantially more mature than long-context or open-ended settings. On ART and e-CARE selection, the strongest models reach 87.2–88.0% accuracy, and diagnosis ranking on DDXPlus reaches 79.75% Top-1 and 98.7% Hit@3 (Table 3). By contrast, longer narrative inference remains harder: the best reported scores are 42.9% on True Detective and 68.0% on the MuSR murder subset. Where human averages are reported, the strongest model remains below them on all four selection tasks with human baselines, and the gap is especially large on MuSR.

Stage I generation is less uniform, both across tasks and across models, as also reflected in the suite-level macro-average in Figure 4. Some benchmarks admit high task-validity or near-saturated performance for the strongest systems, but this is not true across the board. Among the formal missing-premise tasks, AbductionRules is the easiest, with the best model reaching 84.5% accuracy and 96.4 token-level F1, but its exact-match accuracy is not saturated; ProofWriter remains markedly harder, with the best model reaching 21.5% accuracy and 61.6 F1 (Tables 4 and 5). NeuLR also remains nontrivial, although its character-level similarity scores are much higher than its exact accuracy, suggesting that partial or near-miss generations are common.

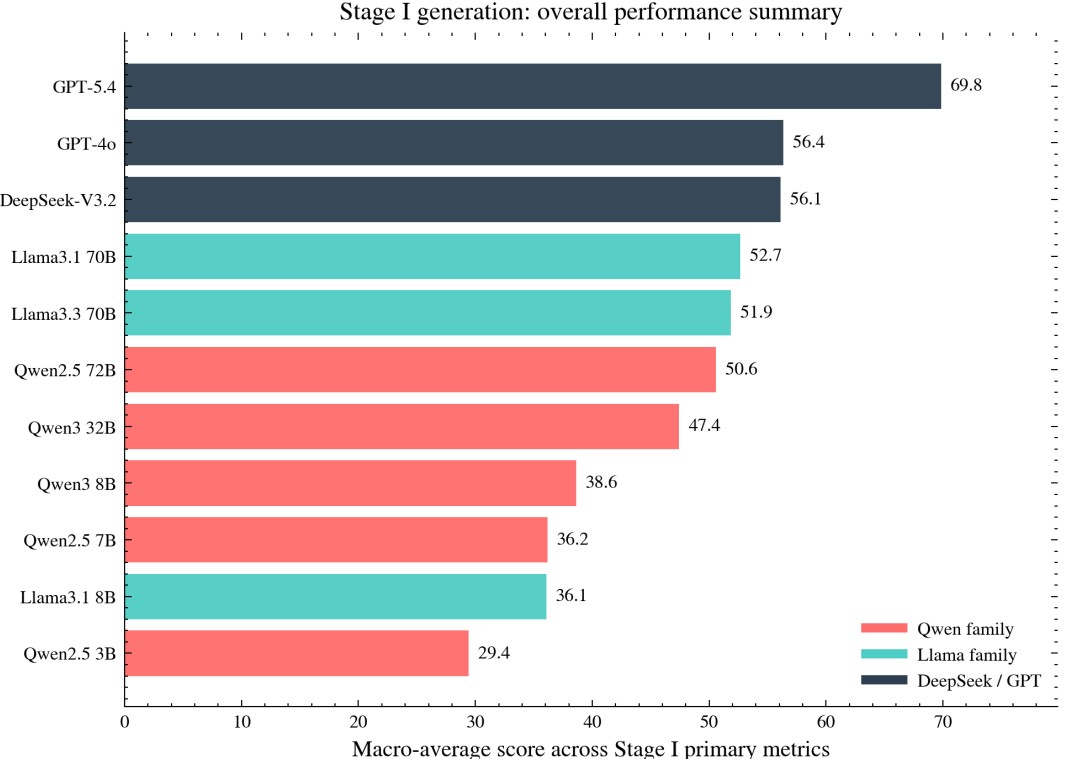

Figure 4: Macro-average performance on the Stage I generation benchmarks. This figure summarizes overall model performance across the evaluated generation tasks.

### Stage I vs. Stage II

One recurring contrast in the suite is between selecting hypotheses and producing them from scratch. The same underlying domain can look much easier once candidate hypotheses are provided. DDXPlus illustrates this especially clearly: in Stage II, the strongest model reaches 79.75% Top-1 and 98.7% Hit@3, whereas the corresponding open-ended diagnosis task reaches 63.7 Hit@3 and 27.9 Set-F1@3. Within this benchmark suite, DDXPlus therefore provides the clearest evidence of a Stage I/Stage II gap, with performance dropping substantially once the model must generate diagnoses rather than rank provided candidates.

For the other paired tasks, the comparison is less decisive. On ART and e-CARE, strong models can often produce valid generations and sometimes compare favorably to human-written references, as the WR-H results in Table 5 indicate. For ART and e-CARE, however, this contrast is harder to interpret cleanly because the generation and selection formulations rely on different evaluation signals, and both tasks may be relatively shallow for the strongest models. A useful next step would be to evaluate a full-pipeline abductive task and analyze where errors arise more often—during hypothesis generation, or during hypothesis selection. That kind of analysis would make it easier to characterize the Stage I/Stage II difference more concretely than aggregate benchmark scores alone allow.

### Domain and Output Structure

The commonsense/expert distinction is useful descriptively, but the difficulty profile in Tables 3–6 does not reduce neatly to that split. DDXPlus, although medically grounded, is relatively tractable when posed as ranking over

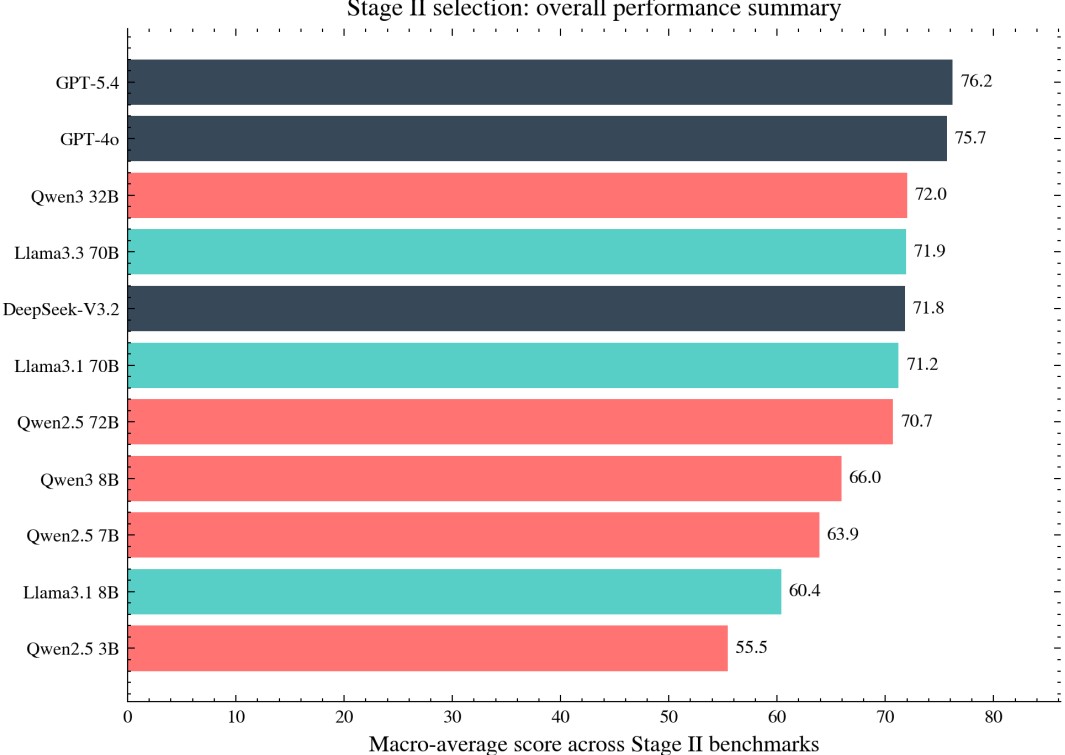

Figure 5: Macro-average performance on the Stage II selection benchmarks. This figure summarizes overall model performance across the evaluated selection tasks.

diagnoses, while UNcommonsense remains challenging despite its everyday setting. Likewise, within the formal family, AbductionRules is comparatively easy because the target is short and tightly constrained, whereas ProofWriter is much harder, likely in part because the task can admit multiple plausible single missing facts rather than a single canonical answer. That makes exact-match scoring demanding and also helps explain why F1 remains relatively low.

Taken together, the results suggest that context length, target structure, and the size of the hypothesis space shape difficulty at least as much as whether the underlying knowledge is commonsense or expert. This is also visible on the selection side: True Detective and MuSR are both natural-language narrative tasks, but they remain materially harder than short two-sentence commonsense bridging benchmarks.

**Metric Sensitivity**

Generation rankings depend noticeably on the metric. On ART, GPT-5.4 obtains the strongest BLEU-4 (4.22), while GPT-4o obtains the strongest ROUGE-L (23.4) and BERTScore (57.0) in Table 6; the highest WR-H in Table 5 belongs to Llama3.1 70B (73.7). On UNcommonsense, DeepSeek-V3.2 has the highest BERTScore, but GPT-5.4 is clearly strongest under the preference-based WR-C and WR-C+L metrics in Tables 4 and 5.

Taken together, these results show that model rankings vary depending on which aspect of abductive generation a metric emphasizes. A model may score highly under lexical or embedding-based similarity yet be less preferred in pairwise judgments, or vice versa. For this reason, no single metric in the benchmark should be treated as a complete proxy for explanatory quality, and performance is best interpreted across multiple complementary views.

**Scale and Model Family Effects**

Within most open-weight families, larger models do better than smaller ones, often by wide margins, as Figure 6 makes clear for the Qwen and Llama series. Qwen2.5 improves steadily from 3B to 72B on nearly every benchmark, and Llama3.1 70B substantially outperforms Llama3.1 8B across both selection and generation. At the same time, performance is shaped not only by parameter count but also by model family and training regime: DeepSeek-V3.2,

despite its much larger parameter count, often does not clearly surpass the strongest Qwen or Llama models in this suite, and GPT-4o still leads GPT-5.4 on some tasks, including MuSR and some generation-side metrics. Overall, larger models tend to perform better, but performance differences across leading models remain benchmark-dependent.

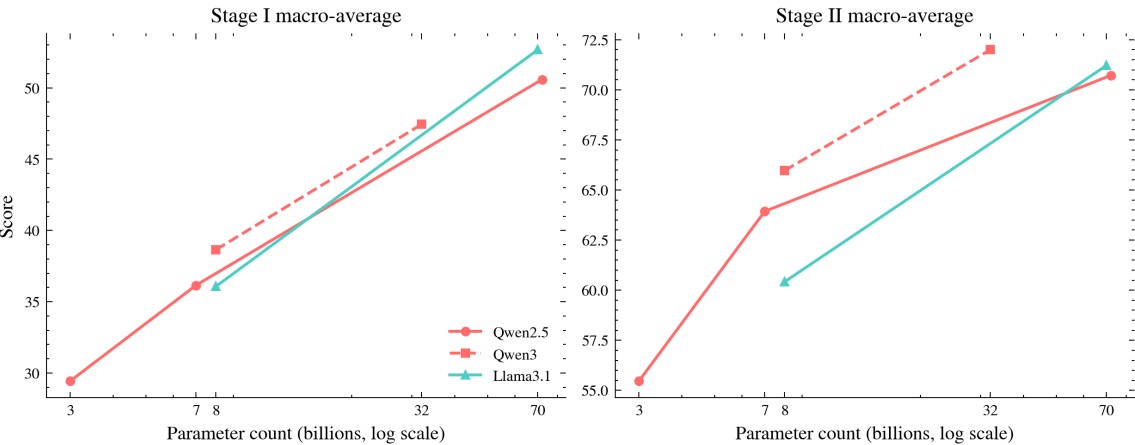

Figure 6: Scaling trends within model families on the Stage I generation and Stage II selection benchmarks. The figure shows that performance generally improves with model scale within the Qwen and Llama families, although the magnitude of improvement varies across families and stages.

### LLM-as-a-Judge Audit

Because several Stage I generation metrics are reference-free, we ran a lightweight judge-model audit to estimate how much the reported results depend on using Gemini 3 Flash Preview as the primary evaluator. For ART, DDXPlus, and UNcommonsense, we sampled 50 examples and re-scored all 11 model outputs with the same judge prompt using Claude Haiku 4.5, yielding 550 paired judge decisions per dataset. Table 7 reports the model-level audit results and includes a pooled row for each dataset.

The audit suggests that the judge choice is not equally load-bearing across tasks. DDXPlus is the most stable case in the displayed audit: the pooled Hit@3 change is only +1.3 points, and the two judges agree on most hit decisions. ART also shows substantial agreement, although Claude is moderately stricter on validity, lowering the pooled validity rate by 6.5 points. UNcommonsense is the most judge-sensitive dataset: Claude assigns noticeably more wins to model generations than Gemini, and direct agreement is lower, consistent with the more open-ended and creative nature of the task. Even there, however, the model-level ranking remains broadly similar across judges, with a Spearman rank correlation of 0.83, so the audit supports the main comparative trends while making clear that absolute judge-based scores, especially for open-ended preference tasks, should be read as judge-dependent estimates rather than judge-independent ground truth.

### Prompting and Decoding Ablations

To test how sensitive Stage I generation results are to the prompting and decoding choices used in the main suite, we ran a diagnostic ablation on 100-example subsets of DDXPlus, UNcommonsense, and ProofWriter. We evaluated three representative models—Qwen3 32B, Llama3.3 70B, and GPT-5.4—under three conditions: the original deterministic Direct Output prompt, the same prompt with temperature 0.7, and a deterministic Chain-of-Thought prompt that asks the model to reason through possible hypotheses before returning a final answer. The evaluation protocol is otherwise kept fixed for each benchmark.

The ablation shows that neither higher-temperature sampling nor Chain-of-Thought prompting uniformly improves Stage I generation. Increasing the temperature produces small and inconsistent changes: it slightly improves DDXPlus Set-F1@3 and improves GPT-5.4 on UNcommonsense WR-C+L, but it does not consistently help the open-weight models and leaves ProofWriter unchanged. Chain-of-Thought prompting has a clearer benefit on more formally constrained generation, raising ProofWriter accuracy for all three models and improving DDXPlus Hit@3 for Qwen3

*ART validity*

| Model | Gemini | Claude | Delta | Agree | Kappa |
|---|---|---|---|---|---|
| Qwen2.5 3B | 64.0 | 48.0 | -16.0 | 84.0 | 0.68 |
| Qwen2.5 7B | 76.0 | 72.0 | -4.0 | 84.0 | 0.59 |
| Qwen2.5 72B | 86.0 | 84.0 | -2.0 | 94.0 | 0.76 |
| Qwen3 8B | 76.0 | 66.0 | -10.0 | 82.0 | 0.57 |
| Qwen3 32B | 84.0 | 76.0 | -8.0 | 84.0 | 0.50 |
| Llama3.1 8B | 74.0 | 74.0 | 0.0 | 84.0 | 0.58 |
| Llama3.1 70B | 90.0 | 90.0 | 0.0 | 96.0 | 0.78 |
| Llama3.3 70B | 92.0 | 80.0 | -12.0 | 88.0 | 0.52 |
| DeepSeek-V3.2 | 92.0 | 84.0 | -8.0 | 92.0 | 0.63 |
| GPT-4o | 88.0 | 86.0 | -2.0 | 86.0 | 0.38 |
| GPT-5.4 | 94.0 | 84.0 | -10.0 | 86.0 | 0.30 |
| **Pooled** | **83.3** | **76.7** | **-6.5** | **87.3** | **0.60** |

*DDXPlus Hit@3*

| Model | Gemini | Claude | Delta | Agree | Kappa |
|---|---|---|---|---|---|
| Qwen2.5 3B | 28.0 | 32.0 | +4.0 | 96.0 | 0.90 |
| Qwen2.5 7B | 34.0 | 34.0 | 0.0 | 96.0 | 0.91 |
| Qwen2.5 72B | 52.0 | 46.0 | -6.0 | 94.0 | 0.88 |
| Qwen3 8B | 50.0 | 50.0 | 0.0 | 96.0 | 0.92 |
| Qwen3 32B | 46.0 | 48.0 | +2.0 | 90.0 | 0.80 |
| Llama3.1 8B | 46.0 | 50.0 | +4.0 | 88.0 | 0.76 |
| Llama3.1 70B | 52.0 | 58.0 | +6.0 | 94.0 | 0.88 |
| Llama3.3 70B | 50.0 | 56.0 | +6.0 | 94.0 | 0.88 |
| DeepSeek-V3.2 | 56.0 | 54.0 | -2.0 | 90.0 | 0.80 |
| GPT-4o | 60.0 | 60.0 | 0.0 | 96.0 | 0.92 |
| GPT-5.4 | 66.0 | 66.0 | 0.0 | 96.0 | 0.91 |
| **Pooled** | **49.1** | **50.4** | **+1.3** | **93.6** | **0.87** |

*UNcommonsense WR-C+L*

| Model | Gemini | Claude | Delta | Agree | Kappa |
|---|---|---|---|---|---|
| Qwen2.5 3B | 8.0 | 22.0 | +14.0 | 82.0 | 0.32 |
| Qwen2.5 7B | 16.0 | 26.0 | +10.0 | 74.0 | 0.23 |
| Qwen2.5 72B | 10.0 | 36.0 | +26.0 | 68.0 | 0.21 |
| Qwen3 8B | 8.0 | 32.0 | +24.0 | 72.0 | 0.26 |
| Qwen3 32B | 30.0 | 46.0 | +16.0 | 72.0 | 0.42 |
| Llama3.1 8B | 14.0 | 40.0 | +26.0 | 66.0 | 0.21 |
| Llama3.1 70B | 34.0 | 50.0 | +16.0 | 60.0 | 0.20 |
| Llama3.3 70B | 46.0 | 48.0 | +2.0 | 74.0 | 0.48 |
| DeepSeek-V3.2 | 30.0 | 62.0 | +32.0 | 48.0 | 0.05 |
| GPT-4o | 26.0 | 50.0 | +24.0 | 56.0 | 0.12 |
| GPT-5.4 | 60.0 | 74.0 | +14.0 | 74.0 | 0.42 |
| **Pooled** | **25.6** | **44.2** | **+18.5** | **67.8** | **0.32** |

Table 7: Judge-model audit on 50 sampled examples per dataset, evaluated across all 11 models. All score, delta, and agreement entries are points on a 0–100 scale. Gemini is the original Gemini 3 Flash Preview score and Claude is the audit score from Claude Haiku 4.5. Delta is Claude minus Gemini. Agreement is direct judge agreement on the corresponding decision: validity for ART, Hit@3 for DDXPlus, and the three-way win/tie/loss judgment for UNcommonsense. Kappa reports Cohen's kappa for the same decision.

| Model | Setting | DDXPlus | | UNcommonsense | | ProofWriter | |
|---|---|---|---|---|---|---|---|
| | | Hit@3 | Set-F1@3 | WR-C+L | WR-C | Acc. | F1 |
| Qwen3 32B | Direct, $T = 0$ | 45.0 | 21.8 | **21.0** | **41.0** | 7.0 | 30.0 |
| Qwen3 32B | Direct, $T = 0.7$ | 46.0 | 23.2 | **21.0** | 39.0 | 7.0 | 30.0 |
| Qwen3 32B | CoT, $T = 0$ | **53.0** | **24.6** | **21.0** | 36.0 | **12.0** | **36.4** |
| Llama3.3 70B | Direct, $T = 0$ | 42.0 | 21.7 | **40.0** | **54.0** | 6.0 | 41.2 |
| Llama3.3 70B | Direct, $T = 0.7$ | 46.0 | 22.3 | 37.0 | 52.0 | 6.0 | 41.2 |
| Llama3.3 70B | CoT, $T = 0$ | **49.0** | **23.2** | 21.0 | 31.0 | **22.0** | **49.6** |
| GPT-5.4 | Direct, $T = 0$ | **63.0** | 27.9 | 59.0 | **82.0** | 30.0 | 65.4 |
| GPT-5.4 | Direct, $T = 0.7$ | 62.0 | **28.9** | **68.0** | 79.0 | 30.0 | 65.4 |
| GPT-5.4 | CoT, $T = 0$ | 59.0 | 26.6 | 57.0 | 75.0 | **35.0** | **69.9** |

Table 8: Prompting and decoding ablations on 100-example Stage I subsets. Direct, $T = 0$ is the deterministic direct-output condition used in the main benchmark suite; Direct, $T = 0.7$ changes only the sampling temperature; CoT, $T = 0$ uses an explicit intermediate-reasoning prompt with deterministic decoding. WR-C+L and WR-C are UNcommonsense win rates against enhanced and original crowd-written references, respectively. All entries are percentages.

32B and Llama3.3 70B. However, it reduces UNcommonsense win rates and does not improve GPT-5.4 on DDX-Plus. Overall, these results suggest that the main Stage I trends are not simply an artifact of greedy decoding: explicit intermediate reasoning can help when the hypothesis space is structurally constrained, but more open-ended explanation generation remains sensitive to prompt format and evaluation signal.

**Interactive SPLAT Diagnostic**

As a complement to the fixed-prompt benchmark suite and the prompting ablation above, we add a small multi-turn diagnostic on SPLAT, a 975-puzzle situation-puzzle benchmark for evaluating lateral thinking in LLMs (Chen et al., 2024). In each puzzle, the model sees an incomplete story, asks yes-or-no questions to a judge with access to the hidden reference scenario, and then submits a final explanation. This makes the setting useful for our purposes because each interaction round contains a local abductive step: the model must maintain a provisional explanation of the hidden scenario, ask a question that constrains it, and revise that explanation as new evidence arrives.

We evaluate Qwen3 32B, Llama3.3 70B, and DeepSeek-V3.2 on 200 proportionally stratified puzzles covering all SPLAT difficulty tiers (45 easy, 133 medium, 22 hard). All models use the same zero-shot Chain-of-Thought player prompt with a maximum of 15 interaction rounds; Gemini 3 Flash Preview is used as the judge. Table 9 reports final solve accuracy.

| Model | Easy | Medium | Hard | Overall |
|-------|------|--------|------|---------|
| Qwen3 32B | 46.7 | 18.0 | 0.0 | 22.5 |
| Llama3.3 70B | **73.3** | **47.4** | **22.7** | **50.5** |
| DeepSeek-V3.2 | 55.6 | 33.1 | 4.5 | 35.0 |

Table 9: SPLAT multi-turn results. Values are final solve accuracy (%) over a 200-example proportional-stratified sample (45 easy, 133 medium, 22 hard).

This diagnostic should be read as complementary rather than conclusive, but it mirrors a pattern visible elsewhere in the benchmark: performance depends strongly on task structure and difficulty. Llama3.3 70B is the strongest of the three models overall, while all models drop substantially from easy to hard puzzles. The sharp drop on hard puzzles suggests that dynamic, multi-step abductive tasks remain difficult, especially when the model must update its hypothesis across several interaction rounds.

## 4.2 Comparative Performance Across Reasoning Types

While the primary focus of this survey is abductive reasoning, understanding how LLM performance on abduction relates to performance on deduction and induction can offer valuable perspective. To this end, we conduct a small-scale empirical comparison by aggregating published results from three recent studies (Sheng et al., 2025; Dougrez-Lewis et al., 2025; Xu et al., 2025a) that evaluate LLMs across multiple reasoning types under comparable conditions.

From these works, we collect accuracy figures spanning a range of representative LLMs evaluated on various datasets covering all three reasoning types. Methods used across these experiments include $k$-shot prompting ($k \geq 0$), chain-of-thought (CoT) prompting, and supervised fine-tuning (SFT). For a complete and detailed list of the specific models, methods, and datasets used in this comparison, we refer the reader to Appendix 9.2.

In the accuracy–distribution plot (Figure 7), each point corresponds to a single experiment consisting of a specific model, prompting/fine-tuning method, and abductive (or deductive/inductive) dataset. In the abduction–vs.–deduction/induction scatter plot (Figure 8), each point corresponds to a *pair* of experiments that share the same model and method: one run on an abductive dataset, and one run on a deductive or inductive dataset.

Figure 7 presents the accuracy distributions across the three reasoning types as box plots. Deduction exhibits the highest median accuracy (79.96%) with a relatively concentrated interquartile range, indicating that current LLMs generally perform well on deductive tasks. In contrast, abduction shows a notably lower median (42.50%) and a wide spread extending down to near-zero accuracy, reflecting both the inherent difficulty of abductive tasks and the substantial variability in how different models and methods cope with them. Induction has the lowest median (28.8%) among the three and also displays a broad variance, indicating considerable variability in performance.

To further examine the relationship between abductive performance and the other reasoning types, Figure 8 presents a scatter plot pairing each configuration's abductive accuracy with its deductive or inductive accuracy on comparable setups. The abduction–deduction pairs (blue) suggest a pattern: the majority of points cluster in a region of low-to-moderate abductive accuracy ($< 50\%$) but high deductive accuracy ($> 60\%$), indicating that strong deductive performance does not consistently imply strong abductive performance. This asymmetry is consistent with the view that abduction requires different capabilities—such as hypothesis generation and plausibility judgment—that are not directly exercised by deductive tasks. The abduction–induction pairs (green), while fewer in number, show a

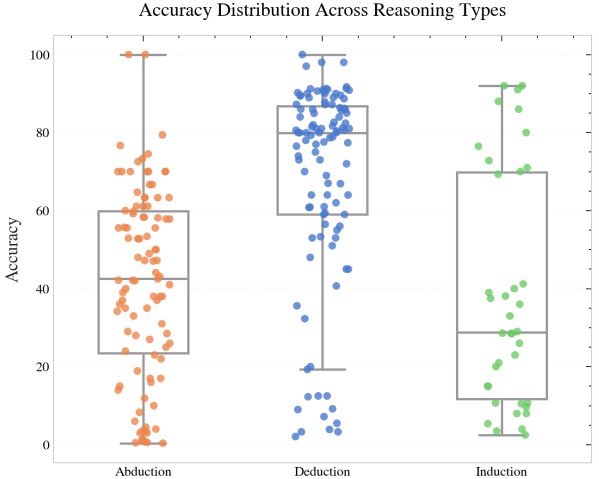

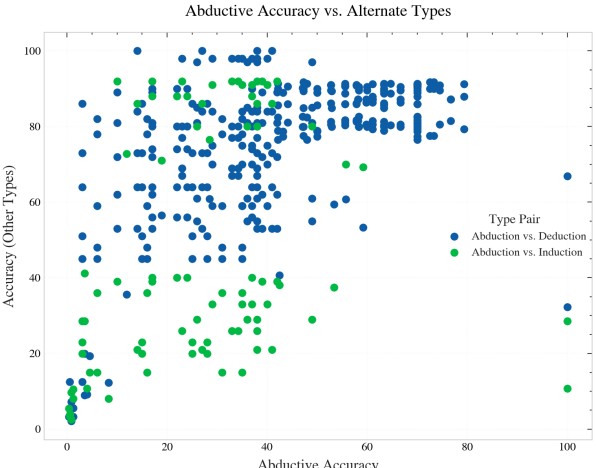

Figure 7: Accuracy distributions across reasoning types. Each point represents a unique (model, method, dataset) combination. This figure represents an aggregated observational comparison compiled from external studies, not a controlled, uniform experiment.

Figure 8: Abductive accuracy plotted against deductive (blue) and inductive (green) accuracy. Each point represents a shared (model, method) configuration evaluated on an abductive dataset and a corresponding deductive or inductive dataset. These pairs represent an observational cross-study comparison rather than a controlled experiment under identical environment conditions.

more dispersed pattern without a clear trend, suggesting that the relationship between abductive and inductive performance is less systematic and likely more dependent on specific task and model characteristics.

We explicitly note that these observations must be interpreted with caution, as this comparison is an observational, post-hoc aggregation of external data rather than a controlled experiment. Consequently, several key conclusions cannot be drawn from this analysis. First, we cannot attribute performance differences solely to the cognitive demands of the respective reasoning types, as confounding variables are not controlled. Second, because the paired tasks do not share the exact same underlying semantic content or structural complexity, we cannot establish a direct, model-specific correlation between deductive/inductive capabilities and abductive capabilities. Finally, given the small sample size of three source papers and the specific selection of models therein, these trends may not generalize universally to all language model architectures or size classes. Ultimately, there is a clear lack of controlled benchmarks that evaluate identical models under completely uniform conditions across abductive, deductive, and inductive paradigms. More systematic, within-paper evaluations utilizing parallelized tasks are needed to provide a more precise understanding of how these distinct reasoning capabilities relate.

# 5 Identified Gaps in Current Research

## 5.1 Lack of Unified Definitions

One of the core challenges in this area is the lack of a shared understanding of what abductive reasoning actually involves. Different works approach it from quite different angles—some frame it purely as a hypothesis selection problem (e.g., (Tchango et al., 2022)), while others treat it as a hypothesis generation task (e.g., (Sheng et al., 2025)). There are also studies that attempt to combine both perspectives, incorporating generation and selection within a single framework (e.g., (Liu et al., 2024)). While each of these directions is valid on its own—largely because, as we have seen in Section 2.1, the concept has always been contentious and stems from fragmented historical roots—the absence of a common definition leads to inconsistencies in how tasks are designed and evaluated. As a result, it becomes difficult to directly compare findings or build systematically on prior work. At a deeper level, this lack of conceptual alignment prevents the field from forming a cohesive structure, and naturally limits its ability to converge toward clear, consistent, and well-established conclusions. This exact gap is the reason we attempt to establish a unified working definition in Section 2.2.1.

## 5.2 Dataset Limitations and the Practical Disconnect

A recurring problem in the current literature is that dataset limitations create two distinct, but related, problems. On one hand, many benchmarks are structurally too shallow and too static to reflect the broader inquiry settings in which abductive hypotheses are actually useful. On the other hand, the domain landscape remains narrower and more fragmented than the motivating applications of abduction would suggest. These two issues should be distinguished: the first concerns *how* abductive reasoning is operationalized, while the second concerns *where* the field is looking for it.

### 5.2.1 Static and Low-Complexity Benchmark Design

The dominant evaluation setting still reduces abduction to a static, one-shot prediction problem. Commonsense benchmarks such as ART/$\alpha$NLI (Bhagavatula et al., 2020) operationalize abduction as selecting a plausible sentence that bridges two observations. Even when the narratives become longer and more difficult, as in True Detective (Del & Fishel, 2023), the task is still ultimately collapsed to a final answer choice, offering limited visibility into the space of candidate hypotheses or the model's basis for preferring one explanation over another.

This simplification also appears in more specialized settings. DDXPlus and L'ART (Tchango et al., 2022; Nguyen et al., 2023) are valuable steps toward domain-specific abductive reasoning, but they still evaluate performance primarily through prediction over a fixed evidential state. Likewise, formal or synthetic settings such as Mini-ARC (Kim et al., 2022) and neuro-symbolic approaches to abductive inference (Aakur & Sarkar, 2023) capture important aspects of explanatory inference, yet they usually remain tightly structured and bounded in ways that make evaluation more tractable than the open-ended abductive reasoning required in realistic settings.

Overall, the limitation is not simply that many benchmarks are easier than real-world abductive tasks. More fundamentally, they compress abduction into a single prediction target under a fixed evidential state, which makes it difficult to assess whether a model can sustain abductive reasoning across more complex or evolving problem settings.

### 5.2.2 Narrow Domain Coverage and Fragmented Framing

A separate limitation concerns domain coverage. The literature remains disproportionately centered on everyday commonsense narratives and a relatively small set of formal or synthetic reasoning tasks (Bhagavatula et al., 2020; Sun & Saparov, 2025). Yet abduction is most often motivated by high-value settings such as complex mathematical reasoning, medical diagnosis, scientific discovery, legal argumentation, engineering and IT, and investigative reasoning. This creates a mismatch between the domains used to justify abductive reasoning and the domains that actually shape benchmark development.

The issue is not only under-coverage, but also fragmentation. Because many real-world tasks are mixed and contain abductive, deductive, and inductive components, a substantial amount of adjacent work studies problems with a strong abductive core without explicitly situating them within the abduction literature. In medicine, related tasks often appear under labels such as *clinical reasoning* or *diagnosis prediction* (Sonoda et al., 2025; Gao et al., 2025; Xu et al., 2024); in scientific discovery, they appear as *biomedical hypothesis generation* (Qi et al., 2024); in narrative reasoning, they appear as *culprit detection* or long-context reasoning (Gupta, 2025; Xu et al., 2025b); and in software engineering, they appear as *code debugging* or *program repair* (Tian et al., 2024). In some cases, this framing difference is especially revealing: WHODUNIT presents culprit identification as a deductive reasoning benchmark, even though, from an abductive perspective, the task also involves inferring the hidden hypothesis that best explains the available clues (Gupta, 2025).

This fragmentation matters because it limits transfer across domains. Datasets, evaluation criteria, and modeling strategies developed in mathematics, medicine, science, law, engineering, and investigative reasoning are rarely compared within a shared abductive framework. Some recent efforts do make the connection explicit—for example, L'ART in legal reasoning and Reasoning Like a Doctor in medical dialogue (Nguyen et al., 2023; Xu et al., 2024)—but these remain exceptions rather than the norm. As a result, the field is missing not just more domains, but also a clearer cross-disciplinary bridge that would allow researchers working on closely related explanatory tasks to recognize that they are often studying different instances of the same underlying reasoning pattern.

## 5.3 Accuracy vs. Genuine Abductive Reasoning

A largely overlooked issue in the literature is the gap between improving task-level accuracy and actually learning abductive reasoning. Although many works report gains on benchmarks, these metrics only assess the end-result

answers to the task, completely bypassing the actual reasoning trace. As a result, it remains unclear whether models truly learn to construct plausible explanatory hypotheses, or simply become better at selecting likely answers.

This concern is supported by recent findings showing that higher accuracy does not necessarily imply better reasoning. For example, Shojaee et al. (2025) demonstrate that models can achieve strong performance on reasoning tasks while relying on superficial patterns rather than genuine inference. In the context of abduction, this is particularly problematic, since the core objective is not just correctness, but the ability to generate and evaluate explanations.

Consequently, focusing solely on accuracy risks overestimating the abductive capabilities of LLMs. A more faithful evaluation should consider the quality of intermediate reasoning—whether generated explanations are coherent, plausible, and truly bridge the gap between observation and hypothesis.

## 5.4 Underexplored Training Paradigms

Supervised Fine-Tuning (SFT) and prompt engineering currently dominate the literature for adapting LMs to abduction. For example, models are routinely trained on benchmark abductive datasets—such as the ART commonsense explanation task (Bhagavatula et al., 2020), the UNcommonsense corpus of unlikely outcomes (Zhao et al., 2024), and the AbductionRules knowledge-base tasks (Young et al., 2022)—to learn patterns of plausible explanations. However, current SFT-based systems are trained merely to imitate reference hypotheses, and likelihood alone is only a weak proxy for whether an explanation is actually good—that is, whether it faithfully accounts for the observations, distinguishes among competing hypotheses, and remains useful beyond the immediate example (Dalal et al., 2024; He et al., 2025a).

Consequently, researchers have begun exploring alternative reinforcement-, imitation-, and preference-based regimes. Reinforcement Learning (RL) for post-training is still largely underexplored in abductive reasoning, and the few early examples focus mainly on structured knowledge-graph settings rather than open-ended natural-language abduction (Bai et al., 2024; Gao et al., 2026b). Existing RL formulations provide a useful starting point but also highlight current limitations. For instance, RLF-KG rewards hypotheses based on how well their execution reconstructs observed facts, using Jaccard-style overlap under PPO (Bai et al., 2024), while CtrlHGen extends this approach with smoother semantic rewards (e.g., Dice and Overlap) and explicit condition-adherence rewards under GRPO (Gao et al., 2026b). These rewards are practical and measurable, but remain relatively narrow: they prioritize reconstruction and control while leaving broader explanatory qualities only weakly specified (Bai et al., 2024; Gao et al., 2026b).

## 5.5 Limited Mechanistic Interpretability of Abductive Reasoning

Despite growing emphasis on mechanistic interpretability in other reasoning domains (Olsson et al., 2022; Maltoni & Ferrara, 2025), research on the interpretability of abductive reasoning processes in LLMs remains in its early stages. While recent work has begun to address interpretability through explicit reasoning traces and structured frameworks (Zheng et al., 2025; He & Chen, 2025; Alkan et al., 2025), deep mechanistic understanding of how models internally perform abduction remains largely unexplored.

Emerging frameworks such as LogiDynamics (Zheng et al., 2025) have made progress in external interpretability by producing structured reasoning traces that enable step-by-step evaluation of hypotheses within multi-agent architectures. Hybrid neuro-symbolic approaches integrate LLMs with symbolic solvers to refine abductive explanations, producing inspectable step-wise reasoning (Kakas & Michael, 2020). Additionally, several works have constructed datasets and benchmarks that provide partial internal interpretability through error analysis and attribution of outputs to specific reasoning flaws (Nguyen et al., 2023; Xu et al., 2025a; He & Lu, 2024).

However, critical mechanistic questions remain largely unanswered: Which specific attention heads activate during hypothesis generation versus evaluation? What circuit-level mechanisms distinguish abductive inference from deductive or inductive reasoning? How do models internally represent and compare competing candidate hypotheses?

This gap is particularly concerning given that abduction inherently involves subjective judgments about explanation quality, making transparency crucial for trust in high-stakes applications such as medical diagnosis and clinical reasoning (Sonoda et al., 2025; Hong et al., 2024; Gao et al., 2025). Without deeper mechanistic interpretability research, we cannot fully verify whether models are performing genuine abductive inference or primarily pattern-matching surface features, nor can we systematically improve their abductive capabilities through targeted circuit-level interventions. The field would benefit significantly from extending the mechanistic interpretability techniques successfully applied to other reasoning domains to the specific challenges of abductive reasoning.

# 6 Future Directions

The gaps identified throughout this survey point to a clear agenda for future work. Progress will require not only better evaluation, but also sharper task formulations, broader domain coverage, and stronger connections between abductive reasoning and the real settings in which explanations are generated, compared, and put to use. In this section, we highlight several directions that we believe are especially important for moving the field beyond narrow benchmark performance and toward more robust abductive reasoning in LLMs.

## 6.1 RL-Based Optimization for Explanatory Virtues

While existing RL approaches provide useful first steps, they remain limited in scope and leave many aspects of abductive quality underspecified. A more mature RL agenda could therefore align reward design more closely with the richer view of explanatory quality developed in this survey. At the most basic level, reward can start from simple, rule-based signals that are easy to compute and verify. These can include basic exact-match–style rewards when a canonical reference hypothesis exists, although such signals are often overly rigid for abduction because multiple hypotheses may be valid, as well as checks on whether a hypothesis entails or reconstructs the observation, satisfies structural constraints, avoids malformed reasoning chains, and whether intermediate steps remain valid under symbolic or verifier-based checks (Bai et al., 2024; Gao et al., 2026b; Guo et al., 2025). Building on these, RL can incorporate more qualitative signals that favor hypotheses which are not only plausible but also explanatory in a stronger sense: coherent, parsimonious, diverse, and capable of supporting informative predictions on unseen inputs (Dalal et al., 2024; He et al., 2025a). Related reasoning work suggests that step-level process rewards may be especially valuable here, since they provide finer credit assignment than outcome-only signals and better capture whether a chain of reasoning stays on track throughout generation and subsequent filtering or selection (Lightman et al., 2024; Wang et al., 2024).

At the same time, recent work on RLHF and related post-training pipelines warns that reward functions for abstract virtues are only proxies susceptible to reward hacking and specification gaming (Rafailov et al., 2024; Mahmoud et al., 2026; Zhao et al., 2025; Williams et al., 2024; Denison et al., 2024). For abductive reasoning, this risk is particularly salient because explanatory virtues are easily imitated superficially: parsimony can collapse into underspecified hypotheses, and coherence into fluent but circular narratives. Therefore, a robust RL program for abduction must combine virtue-oriented rewards with strong safeguards. Promising directions include using multi-signal objectives that blend hard-to-fake constraints (e.g., consistency with the observation, factual correctness, and symbolic validity) with process-level rewards and held-out evaluators (Lightman et al., 2024; Wang et al., 2024; Mahmoud et al., 2026). To mitigate reward tampering, optimization can be stabilized via reference-policy regularization (Laidlaw et al., 2025), gradient regularization (Ackermann et al., 2026), adversarial reward training (Bukharin et al., 2025; Zhao et al., 2025), and causal reward modeling to reduce spurious correlations like verbosity (Wang et al., 2025). Ultimately, progress must be evaluated not merely by reward gains, but by whether improvements survive evaluator shifts, adversarial perturbations, expert review, and downstream tests of explanatory usefulness.

## 6.2 Toward Richer and Broader Abductive Benchmarks

If current datasets are limited both by benchmark structure and by domain coverage, future work can address both problems directly. The next generation of abductive resources can feature more complex and conceptually challenging reasoning while drawing from a broader range of communities and applications. These two directions are complementary: richer benchmark design can better evaluate the individual abductive steps that arise in realistic reasoning pipelines, while broader domain coverage makes the field more relevant and more connected to adjacent literatures already working on explanatory reasoning.

### 6.2.1 Richer Structure and Action-Oriented Evaluation

A natural next step is to move beyond static, one-shot datasets toward benchmarks that treat abduction as more than a single prediction target. Future resources should evaluate not only whether models can generate or select a plausible explanation, but also whether they can compare alternatives, justify a choice, and support downstream reasoning or decision-making. This does not require redefining abduction as inherently iterative. Rather, it requires benchmarks that better capture how abductive steps function within broader multi-step settings: clinicians narrow differentials over time, investigators revise theories as evidence accumulates, and engineers trace failures through layered causal structure. Benchmarks should reflect this by exposing partial evidence, intermediate latent structure, and competing explanations whose plausibility depends on context.

**Reasoning-Trace Data Synthesis.** One promising route is to construct richer abductive data from existing structured reasoning traces. Related ideas already appear in datasets such as ProofWriter and AbductionRules, which operationalize abduction through controlled missing-premise or missing-explanation settings (Tafjord et al., 2021; Young et al., 2022). Formal proofs, code execution traces, scientific simulations, and similar artifacts already contain dependency structure that can be repurposed for more challenging benchmark design. If a valid trace is written as $D(P \cup R) \rightarrow C$, dataset construction can hide $P$ or part of the intermediate chain and ask the model to recover a premise set $P'$ that makes $C$ intelligible under $R$. The broader opportunity is to extend this paradigm beyond tightly controlled logical settings toward expert-grounded benchmarks in areas such as coding, mathematics, medicine, and scientific reasoning.

**Action-Oriented Evaluation.** Greater structural richness alone is not enough. Abduction matters not only because it yields plausible explanations, but because those explanations often guide what should happen next. Evaluation should therefore become more **action-oriented**: hypotheses should be judged by whether they support the right next diagnostic test in medicine, the right evidence request in legal or investigative settings, the right experiment in scientific discovery, or the right localization or patching step in debugging. More broadly, benchmarks can move toward interactive pipeline settings in which abduction is assessed through its contribution to subsequent reasoning or action. In such settings, explanation quality is measured not only by plausibility, but also by how well it narrows the hypothesis space, supports the next decision, and improves downstream task success.

### 6.2.2 Exploring Abductive Reasoning Across Diverse Domains

Future progress depends on actively exploring and studying abduction in domains where inferring the best explanation is already central to human expertise. Rather than treating abduction as a niche problem within commonsense NLI, the field should investigate disciplines where reasoning inherently works backward from observations to likely causes or missing links. By placing these domains into a shared framework, ideas about modeling and evaluation can transfer across communities. Notable domains ripe for abductive exploration include:

- **Complex Mathematical Reasoning:** Identifying a necessary auxiliary line or lemma is a form of creative abduction, where solvers guess the missing link to complete a proof.

- **Medicine and Healthcare:** Medical diagnosis is fundamentally abductive; practitioners observe symptoms and infer the most plausible underlying disease to initiate treatment without absolute certainty.

- **Law and Forensics:** Crime scene investigators and juries piece together scattered evidence to construct the most plausible timeline, narrative, or suspect identity.

- **Engineering and IT:** Troubleshooting software bugs and diagnosing mechanical failures require hypothesizing root causes from surface-level system errors or traffic anomalies.

- **Science and Research:** Formulating hypotheses for novel phenomena (e.g., a dip in a star's brightness) or inferring the culture of lost civilizations from archaeological fragments rely heavily on proposing best-fit explanations for incomplete data.

This does not mean simply relabeling every reasoning problem as abduction. Many real-world tasks are mixed and contain abductive, deductive, and inductive components. The more useful goal is explicit decomposition. Future research would benefit from clarifying which part of a domain's pipeline involves hypothesis generation, hypothesis selection, or verification. Such clarification makes it possible to study other related domains' tasks within a shared framework without pretending they are identical. Abductive reasoning research can greatly benefit from the richer tasks and stronger domain grounding of these applied fields. Making these links explicit will reduce duplication across domains and help abductive reasoning mature from a narrow evaluation category into a genuinely cross-disciplinary research program.

### 6.3 Generalized Multi-Agentic Abductive Frameworks

Recent studies suggest that structured multi-agent interaction—particularly through mechanisms such as debate and discussion—can enhance reasoning by promoting divergent thinking and improving properties like diversity and originality in generated outputs (Liang et al., 2024; Lu et al., 2024; Chen et al., 2025). These findings are especially relevant to abductive reasoning, which fundamentally relies on generating, comparing, and selecting among competing explanatory hypotheses. This connection suggests that distributing the abductive process across multiple interacting agents could, in principle, yield richer hypothesis spaces and more robust evaluation dynamics than single-agent approaches.

However, while multi-agent systems have proven effective for abductive reasoning in specific, narrow domains—such as medical diagnostics (Hong et al., 2024)—they have yet to be broadened into a universal tool. Currently, there is no generalized, domain-agnostic multi-agent framework explicitly designed to handle the abductive reasoning process across arbitrary tasks. Given the clear success of multi-agent dynamics in specialized fields, systematically testing and developing a universal multi-agent architecture for general abduction presents a highly promising, yet largely unexplored, avenue for research.

### 6.4 Advancing Mechanistic Interpretability of Abductive Circuits

While recent work has begun making abductive reasoning more transparent through explicit reasoning traces (Zheng et al., 2025; He & Chen, 2025; Alkan et al., 2025) and hybrid neuro-symbolic approaches (Kakas & Michael, 2020), a critical frontier remains: understanding *how* models perform abduction at the circuit and component level. Building on emerging interpretability research (Liu et al., 2025; Yang et al., 2023; Wu et al., 2025a), we propose a research agenda centered on identifying and manipulating the neural mechanisms responsible for abductive inference.

**Circuit Discovery and Localization**  Using techniques from mechanistic interpretability such as activation patching, causal tracing, and sparse probing, future work should systematically identify which transformer components (attention heads, MLP layers, residual streams) activate specifically during abductive sub-processes. While general attention patterns in LLMs are increasingly understood ( (Conmy et al., 2023),  (Kramár et al., 2024)), their specific role in abductive reasoning requires dedicated investigation. Key questions may include: How are candidate hypotheses represented and compared in the model's latent space? Which specific attention heads activate during hypothesis generation versus evaluation?

**Comparative Circuit Analysis Across Reasoning Types**  A critical research direction is differentiating abductive circuits from deductive and inductive ones. This comparative analysis would reveal whether abduction is a compositional reuse of deductive/inductive primitives or involves fundamentally distinct mechanisms—a question that existing error analysis and performance evaluation studies cannot answer (Hong et al., 2025).

**Targeted Circuit Intervention and Enhancement**  Once abductive circuits are identified, researchers can test causal hypotheses through ablation studies (removing or dampening specific components) and steering interventions (amplifying activations in identified circuits)  (Meng et al., 2022), (Bayat et al., 2025). For instance, if a specific attention head pattern correlates with selecting more general hypotheses, artificially enhancing its activation should bias the model toward that goal. Such interventions could enable fine-grained control over selection preferences without requiring full model retraining.

## 7 Conclusion

In this survey, we examined abductive reasoning as a distinctive mode of reasoning and investigated its emerging role within large language models. Given the absence of a universally accepted definition of abduction in LLMs, we conducted a historical and conceptual analysis of the notion of abduction, tracing its development from its philosophical origins to its modern interpretations. Building on this foundation, we established a working definition of abductive reasoning tailored to LLMs, aiming to provide conceptual clarity for future research in this area.

With this definition in place, we proposed a structured categorization of existing work on abductive reasoning in LLMs. We reviewed and classified the body of published research according to task formulations, types of datasets used, methodological approaches, and evaluation paradigms to provide a compact view of prior work and to show how different studies relate to one another across the survey.

To empirically ground the framework, we conducted a benchmark study evaluating a range of open- and closed-source LLMs on standard abductive benchmarks. This included targeted comparative analyses across model sizes, model families, evaluation types, and the distinct generation-versus-selection settings implied by our two-stage definition. By synthesizing recent empirical results, we further examined how performance on abductive tasks relates to performance on deductive and inductive benchmarks, offering a comparative view of current models' broader reasoning profiles.

Our findings show that, despite impressive progress in general-purpose language modeling, abductive reasoning in LLMs remains at an early stage. Persistent gaps span multiple levels: conceptually, there is still no widely adopted framework for defining and measuring abduction; empirically, benchmarks remain largely static, domain-narrow, and insufficiently sensitive to the two-stage nature of abductive reasoning; methodologically, training regimes rarely target explanatory quality or model uncertainty over hypotheses; and mechanistically, we have limited understanding of how

internal circuits implement hypothesis generation and selection, or how these processes interact with other reasoning modes.

Addressing these gaps calls for coordinated advances along several fronts. On the evaluation side, richer, dynamic, and action-oriented benchmarks are needed—benchmarks that stress-test both generation and selection, span broader domains, and connect explanations to downstream consequences. On the algorithmic side, reinforcement learning and related techniques could be used to align models with explanatory virtues such as coherence, parsimony, and robustness under new evidence. Multi-agent and modular architectures offer promising avenues for decomposing abductive processes into interacting components specialized for hypothesis proposal, critique, and revision. Finally, progress in circuit-level interpretability and mechanistic analysis will be crucial for uncovering how abductive inferences are internally represented and manipulated.

Taken together, we hope that the framework, evaluation, and synthesis presented in this survey provide a solid foundation for advancing the study of abductive reasoning in large language models and encourage deeper engagement with this underexplored yet essential dimension of machine reasoning.

# 8 Broader Impact Statement

While this survey focuses on the theoretical and empirical landscape of abductive reasoning in Large Language Models (LLMs), the practical deployment of these models carries broader social and safety implications. First, abductive reasoning is a key component in complex decision-making domains such as medicine, legal analysis, and scientific research. The recognized gap in the literature between high benchmark performance and genuine, robust abductive inference highlights a potential risk: models relying on superficial pattern matching rather than valid explanatory logic may generate plausible-sounding but incorrect hypotheses. In high-stakes environments, such errors could lead to incorrect diagnostic or analytical conclusions, emphasizing the necessity of human oversight and verification rather than sole reliance on model outputs.

Second, proposed research directions regarding circuit-level interpretability and model steering introduce potential dual-use concerns. While these techniques are designed to verify reasoning pathways and improve model reliability, methods that steer internal activations can also be used to bias the generation or selection of hypotheses. In sensitive contexts, this capability could be used to intentionally skew generated explanations toward specific narratives. Consequently, research into abductive capabilities and steering mechanisms should be accompanied by the development of corresponding safeguards, transparency frameworks, and robust evaluation protocols to mitigate risks of systematic bias or misuse.

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

# 9 Appendix

## 9.1 Benchmark Construction and Evaluation Details

This appendix documents the concrete benchmark construction, prompting, and evaluation protocol behind Section 4.1. The suite is intentionally selective rather than exhaustive. Its purpose is to instantiate both stages of our working definition, span both dataset families discussed in Section 3.2, and cover both closed-form and open-ended evaluation under a reasonably uniform experimental setup.

Across the main benchmark experiments, we used one task-specific direct-instruction prompt per benchmark formulation and kept that prompt fixed across models. For the main benchmark results in Tables 3–6, we used deterministic decoding (temperature 0); the diagnostic ablation in Table 8 deliberately varies prompting and temperature. API-based models were queried through OpenRouter, while Qwen2.5-3B was run locally in 16-bit precision. For each benchmark, the evaluated sample set was held fixed across models to keep comparisons consistent. Outputs were constrained as tightly as possible to the benchmark-native answer format (option index, ranked diagnosis list, short explanation, or controlled missing fact(s)). Judge-based metrics were used only where exact matching or reference overlap would clearly understate performance; in our setup, all such judge calls used Gemini 3 Flash (preview). Because some experiments use fixed subsets, specific benchmark splits, or task formulations that differ from those in the original benchmark papers, the reported numbers should be interpreted with these evaluation choices in mind when comparing them with prior results.

On the selection side, the suite combines short commonsense bridging, medical diagnosis ranking, and longer narrative mystery solving. On the generation side, it combines open-text explanation, open-ended diagnosis generation, and formally constrained missing-premise completion. Tables 10 and 11 summarize the exact subsets and task formulations used in this study.

| Benchmark | Subset used in this study | Stage II formulation | Primary metrics |
|---|---|---|---|
| ART | 500 examples | Two-observation abductive selection with two candidate hypotheses; output only 1 or 2. | Accuracy |
| e-CARE | 500 examples from the *cause* split only | Event-to-cause selection with two candidate causes; the *effect* split is omitted because it does not instantiate the abductive direction targeted here. | Accuracy |
| DDXPlus | 400 examples | Rank the provided candidate diagnoses from most to least likely; output diagnosis numbers only. | Top-1, Hit@3 |
| True Detective | 191 cases (entire dataset) | Solve the mystery from a long story and choose the single best answer option. | Accuracy |
| MuSR (murder) | 250 cases (entire murder split) | Read a short murder mystery and select the most likely culprit. | Accuracy |

Table 10: Stage II (hypothesis selection) benchmarks used in Section 4.1.

### Formulation Choices

Three paired benchmarks—ART, e-CARE, and DDXPlus—appear in both stages. This is deliberate: they let us compare selection and generation while holding the underlying domain fixed. In ART, the difference is whether the model chooses between two candidate bridges or writes one directly. In e-CARE, the difference is whether the model selects the more plausible cause or produces the conceptual statement that makes the cause–effect relation intelligible. In DDXPlus, the difference is especially informative because the open-ended version removes the benchmark's candidate set entirely and therefore tests whether the model can surface plausible diagnoses rather than merely rank them.

The remaining benchmarks were chosen to cover abductive settings that the paired tasks do not. True Detective and the murder subset of MuSR stress longer narrative contexts with dispersed evidence and explicit culprit selection. UNcommonsense isolates open-ended commonsense explanation in a many-to-one setting where multiple explanations may be reasonable. ProofWriter, AbductionRules, and NeuLR cover the formal end of the space: here the model must generate missing information under explicit structural constraints, and evaluation can therefore combine exact correctness with overlap-sensitive secondary metrics.

| Benchmark | Subset used in this study | Stage I formulation | Primary metrics |
|---|---|---|---|
| ART | 400 examples | Generate one short intermediate event or state that bridges two observations. | Val., WR-H, BLEU-4, ROUGE-L, BERTScore |
| e-CARE | 400 examples from the *cause* split | Generate one short conceptual explanation for an explicit cause–effect pair derived from the abductive *cause* setting. | Val., CEQ, BLEU, ROUGE-L |
| DDXPlus | 400 examples | Generate exactly three likely diagnoses directly, without a candidate list; this is our own open-ended formulation of the benchmark. | Hit@3, Set-F1@3 |
| UNcommonsense | 200 examples, with source proportions matched to the original dataset composition | Generate a plausible explanation for an unlikely outcome; the prompt template is selected by source (un-SocialIQA or un-RocStories). | WR-C, WR-C+L, BERTScore |
| ProofWriter | 400 examples from the D5-Ab test set, with 80 examples from each depth level 1–5 | Output all single missing facts that would each individually make the query provable. | Accuracy, F1 |
| AbductionRules | 400 stratified examples, sampled equally from *person simple*, *human simple*, *person*, and *human* | Output exactly one missing fact that makes the query decidable, i.e., provable or disprovable. | Accuracy, Tok-F1 |
| NeuLR | 250 examples from the abductive split | Output the single missing fact that makes the target fact derivable from the context. | Accuracy, CLS. |

Table 11: Stage I (hypothesis generation) benchmarks used in Section 4.1.

## Model Snapshots and Restricted Subsets

To make the benchmark runs reproducible, Table 12 reports the exact model identifier, snapshot or release information, and access method used for each model in Tables 3–6. Unless otherwise noted, models were accessed through the OpenRouter chat-completions API, with the `model` field set to the identifier shown in the table. We did not use OpenRouter router aliases such as `:nitro`, `:floor`, `:free`, or `~openai/gpt-latest`, and we did not set provider-specific routing constraints in the benchmark scripts; requests therefore used OpenRouter's default routing for the selected model identifier. Qwen2.5-3B-Instruct was the only model run outside OpenRouter: it was loaded locally on Google Colab with Hugging Face Transformers in 16-bit precision.

Restricted subsets were drawn once before model evaluation by uniform sampling without replacement from the stated official split unless Tables 10–11 specify a stratification constraint; the sampling scripts, fixed seed, and exact example identifiers are available in this repository: `https://github.com/ANONYMOUS/abductive-reasoning-splits` (this will be replaced with the real project URL upon acceptance.)

Table 12: Model identifiers, snapshot information, and access methods used in the benchmark. OpenRouter rows report the identifier used in the request body.

| Model label in tables | Model identifier / snapshot | Release or snapshot information | Access method |
|---|---|---|---|
| Qwen2.5 3B | `Qwen/Qwen2.5-3B-Instruct` | Qwen2.5 family release: 2024-09-19; Hugging Face revision: `aa8e725` ,(main, last modified 2024-09-25) | Local Colab inference, fp16 |
| Qwen2.5 7B | `qwen/qwen-2.5-7b-instruct` | OpenRouter release date: 2024-10-16; undated OpenRouter model slug | OpenRouter API |
| Qwen2.5 72B | `qwen/qwen-2.5-72b-instruct` | OpenRouter release date: 2024-09-19; undated OpenRouter model slug | OpenRouter API |
| Qwen3 8B | `qwen/qwen3-8b-04-28` | Dated OpenRouter identifier; release date: 2025-04-28 | OpenRouter API |
| Qwen3 32B | `qwen/qwen3-32b-04-28` | Dated OpenRouter identifier; release date: 2025-04-28 | OpenRouter API |
| Llama3.1 8B | `meta-llama/llama-3.1-8b-instruct` | OpenRouter release date: 2024-07-23; undated OpenRouter model slug | OpenRouter API |

*Continued on next page*

| Model label in tables | Model identifier / snapshot | Release or snapshot information | Access method |
|---|---|---|---|
| Llama3.1 70B | `meta-llama/llama-3.1-70b-instruct` | OpenRouter release date: 2024-07-23; undated OpenRouter model slug | OpenRouter API |
| Llama3.3 70B | `meta-llama/llama-3.3-70b-instruct` | OpenRouter release date: 2024-12-06; undated OpenRouter model slug | OpenRouter API |
| DeepSeek V3.2 | `deepseek/deepseek-v3.2-20251201` | Dated OpenRouter identifier; release date: 2025-12-01 | OpenRouter API |
| GPT-4o | `openai/gpt-4o` | OpenRouter release date: 2024-05-13; undated OpenRouter model slug | OpenRouter API |
| GPT-5.4 | `openai/gpt-5.4-20260305` | Dated OpenRouter identifier; release date: 2026-03-05 | OpenRouter API |
| Gemini 3 Flash Preview | `google/gemini-3-flash-preview-20251217` | Dated OpenRouter preview snapshot: 2025-12-17 | OpenRouter API |

## Exact Prompting Templates

Below we report the task prompts used in our experiments, with dataset-specific fields shown in standardized placeholder form.

### P1. ART Selection

```
You are given two observations and two possible hypotheses.
Choose the hypothesis that best explains the observations.

Observation 1:  {observation_1}
Observation 2:  {observation_2}

Hypothesis 1:  {hypothesis_1}
Hypothesis 2:  {hypothesis_2}

Answer with only:  1 or 2.
```

### P2. e-CARE Selection

```
You are given an event and two candidate causes.
Choose the more plausible cause of the event.

Event:  {event}

Option 1:  {option_1}
Option 2:  {option_2}

Answer with only:  1 or 2.
```

### P3. DDXPlus Selection

```
You are given a patient case and a list of candidate diagnoses.
Provide a ranked differential diagnosis from most likely to least likely.
Use only the provided diagnosis options.
Reply with the diagnosis numbers only, one per line, in ranked order.
Do not add any extra text.
Example:
1
4
2

Patient case:
{clinical_summary}

Candidate diagnoses:
{candidate_diagnoses_numbered}

Answer:
```

### P4. True Detective Selection

```
You are given a mystery name, a mystery story, and a list of answer options.
Choose the answer option that best solves the mystery.
Answer with only the number of the correct option.

Mystery name:  {case_name}
Mystery story:  {mystery_text}

Answer options:
{options_numbered}

Answer:
```

### P5. MuSR Selection

```
Read the murder mystery and answer the question by selecting the most likely option.

Scenario:
{narrative}

Question:  {question}

Options:
{options_numbered}

Return only the number of the best option.

Answer with only one of:  {valid_labels}.
```

### P6. ART Generation

```
You are given two observations from a short narrative.

Observation 1:  {observation_1}
Observation 2:  {observation_2}

Write the missing abductive explanation:  one short sentence that could naturally happen between the two
observations.

Requirements:
- write exactly one sentence
- output only the missing intermediate event or state
- keep it short, plain, and concrete
- make it plausible with both observations
- do not restate Observation 1 or Observation 2
- do not add extra causes, background details, or consequences
- do not explain why
- do not use introductory phrases or commentary

Aim for a brief bridge like the reference target, not a full explanation.

Output only the missing explanation.
```

### P7. e-CARE Generation

```
You are given a causal fact.

Cause:  {cause}
Effect:  {effect}

Write the missing conceptual explanation as a short factual statement.
The explanation should name the underlying property, rule, or definition.
Do not explain your reasoning.
Do not use phrases like:
- "The principle..."
- "the mechanism..."
- "this causal relation..."
- "because"
- "therefore"

Style requirements:
- one sentence only
- plain factual wording
- as short as possible
- no introductory words
- no references to the cause/effect pair
- no extra commentary

Output only the conceptual explanation.
```

**P8. DDXPlus Generation**

```
You are given a patient case.
Provide the 3 most likely diagnoses.
Reply with exactly 3 lines, one diagnosis per line, in no specific order.
Use this format only:
1.  diagnosis 1
2.  diagnosis 2
3.  diagnosis 3
Each line must contain only the diagnosis name after the number.
Do not add any extra text, commentary, or reasoning.

Patient case:
{clinical_summary}

Answer:
```

**P9a. UNcommonsense Shared Instruction**

```
You are writing an abductive explanation for an unexpected outcome.
Write a plausible explanation that naturally follows the context, makes the outcome more likely, and leaves
as little information gap as possible.
Write only the explanation.
Do not restate the context or the outcome.
Use 1 to 3 sentences.
```

**P9b. UNcommonsense User Template: un-SocialIQA**

```
Context:  Cameron decided to have a barbecue and gathered her friends together.
Outcome:  Others feel bored and uninterested.
Explanation of the outcome:  Other than eating the food, there weren't other activities planned.

Context:  Jan needed to give out jobs for an upcoming project at work.
Outcome:  Others will take a nap instead of working.
Explanation of the outcome:  The others don't get paid more for doing the jobs Jan gave out.

Context:  Remy was an expert fisherman and was on the water with Kai.  Remy baited Kai's hook.
Outcome:  Remy will eat a sandwich.
Explanation of the outcome:  It's been too long before they feel the weight of a fish, and Remy is hungry.

Context:  {context}
Outcome:  {outcome}
Explanation of the outcome:
```

**P9c. UNcommonsense User Template: un-RocStories**

```
Context:  My friends all love to go to the club to dance.  They think it's a lot of fun and always invite.
I finally decided to tag along last Saturday.  I danced terribly and broke a friend's toe.
Outcome:  My friends decided to keep inviting me out as I am so much fun.
Explanation of the outcome:  My friends thought the way I dance is really funny and they couldn't stop
laughing.

Context:  On the fourth of July, Lilly baked a lemon blueberry cake.  She brought it to her boyfriend's
house and they had a bbq.  After dinner they drove into the city to watch fireworks.  When the show was
over, they got donuts from a food truck.
Outcome:  Lilly had a terrible date.
Explanation of the outcome:  Lilly's boyfriend kept complaining that the donuts were way better than the
lemon blueberry cake she made, and her boyfriend just threw the cake away.

Context:  Jennifer was bored one Saturday.  She decided to alleviate her boredom with a hike.  She drove to
a national park to go hiking.  Jennifer hiked for hours.
Outcome:  Jennifer thought hiking was stupid.
Explanation of the outcome:  She realized the Saturday was a holiday, and the hiking trails in the national
park were too crowded that it took her longer than usual to finish.

Context:  {context}
Outcome:  {outcome}
Explanation of the outcome:
```

**P10. ProofWriter Generation**

```
You are given a theory and a query.

The query is not currently provable from the theory.

Your task is to output all single missing facts that, if added individually to the theory, would make the
query provable.

Rules:
- Output only the missing facts.
- Each missing fact must be a single fact, not a rule.
- If there are multiple valid answers, output all of them.
- Put each answer on its own line.
- If there is no valid single missing fact, output exactly:
None
- Do not explain your reasoning.

Theory:
{theory}

Query:
{query}

Answer:
```

### P11. AbductionRules Generation

```
You are given a context containing facts and rules and an observation from the AbductionRules task.

Your task is to output the single missing fact, called the explanation or label, that makes the observation
logically decidable from the context:  either provable or disprovable when the fact is added.

Important:
- The observation may be affirmative or negated.
- The correct fact may help prove the observation, or help derive the opposite and thereby disprove it.
- Choose a direct, minimal fact about an entity or entities in the context.
- Do not output a rule.
- Do not use external knowledge.
- Keep the wording as close as possible to the facts already used in the context.
- Output exactly one fact as one sentence ending with a period.
- Output only the fact.
- Do not explain.
- Do not output a list.
- Do not write anything before or after the fact.

Context:
{THEORY}

Observation:
{QUERY}

Explanation:
```

### P12. NeuLR Generation

```
You are given a context and a target fact.

The target fact is not derivable from the current context.

Output the single missing fact that, if added to the context, would make the target fact derivable.

Rules:
- Output exactly one missing fact.
- The answer must be a single fact, not a rule.
- Output only the missing fact.
- Do not explain your reasoning.

Context:
{context}

Target fact:
{target_fact}

Answer:
```

### Judge Prompts and Matching Rules

Judge prompts were used only when exact string matching would clearly understate performance or fail to capture explanatory adequacy. The natural-language prompts below are reproduced exactly up to placeholder normalization.

### J1. ART Validity Judge

```
You are evaluating an abductive explanation between two observations.

Consider these qualities:
- consistency with Observation 1 and Observation 2,
- plausibility as a bridge between them,
- whether it gives a meaningful explanation,
- clarity and minimality.

Observation 1:  {observation_1}
Observation 2:  {observation_2}
Candidate explanation:  {candidate}

Is the candidate a valid explanation connecting the two observations?

Reply with exactly one label on the first line:
VALID
or
INVALID
```

### J2. ART Pairwise Judge (for WR-H)

```
You are comparing two abductive explanations for the same pair of observations.

Choose which explanation is better overall.
Consider these qualities:
- consistency with both observations,
- plausibility,
- explanatory adequacy,
- clarity and minimality.

Observation 1:  {observation_1}
Observation 2:  {observation_2}

Explanation A: {candidate_a}
Explanation B: {candidate_b}

Reply with exactly one label on the first line:
A
B
EQUALLY_GOOD
EQUALLY_BAD
```

### J3. e-CARE Validity Judge

```
You are evaluating whether a generated conceptual explanation can explain a causal fact.

Causal fact:
Cause:  {cause}
Effect:  {effect}

Generated explanation:
{prediction}

Task:
Decide whether the generated explanation can explain why the cause can lead to the effect.

Judging standard:
* Mark VALID if the explanation states a plausible general principle, property, mechanism, or condition that
makes the causal relation understandable.
* Mark INVALID if the explanation is irrelevant, contradictory, factually wrong, too vague to explain the
relation, or merely restates the cause/effect without giving the underlying principle.
* Prefer judging the explanatory adequacy of the generated explanation itself, not whether it matches any
specific reference wording.
* Be strict but fair.

Return JSON only with this schema:
{
"label":  "VALID" or "INVALID",
"confidence":  0.0 to 1.0,
"reason":  "one short sentence"
}
```

### J4. DDXPlus Strict Disease-Matching Judge

```
Match diseases between the two lists.
Match only if they are 100% the same diagnosis.
Allow exact synonyms, spelling variants, abbreviation expansion, and singular/plural variants.
Do not match broader or narrower diseases, complications, related syndromes, or clinically similar
conditions.
Use one-to-one matching only.

Gold list (0-indexed):
{gold_list_numbered}

Predicted list (0-indexed):
{pred_list_numbered}
```

In DDXPlus generation, the judge response was additionally constrained with a strict JSON schema implementing one-to-one matching between gold and predicted diagnoses. We omit that schema here because it is an implementation detail rather than a natural-language prompt.

**J5. UNcommonsense Pairwise Judge**

```
Choose which explanation better makes the uncommon outcome less surprising given the context.

Prefer the explanation that better fits the context, makes the outcome more likely, adds the missing link,
and leaves little information gap.  Prefer context-specific explanations over generic ones.

Do not prefer length, detail, fluency, polish, or AI-like style.
Do not reward restating the outcome.
Penalize contradiction, irrelevance, implausibility, and vagueness.

Context:
{context}

Outcome:
{outcome}

Explanation A:
{answer_a}

Explanation B:
{answer_b}

Return JSON only:
{
"winner":  "A" | "B" | "tie_good" | "tie_bad",
"reason":  "one short sentence"
}
```

**Metric Glossary**

Table 13 defines the abbreviated metrics reported in Section 4.1. We separate primary task metrics from complementary metrics in the main text, but collect the operational definitions here for convenience.

Table 13: Metric glossary for Section 4.1.

| Metric | Used on | Operational definition in this study |
| --- | --- | --- |
| Accuracy | ART, e-CARE, True Detective, MuSR, ProofWriter, AbductionRules, NeuLR | Exact correctness under the task's native answer key or verification procedure. On ProofWriter, this is exact-set accuracy: the full predicted set of missing facts must match the gold set. |
| Top-1 | DDXPlus selection | Whether the gold primary diagnosis is ranked first among the provided candidate diagnoses. |
| Hit@3 | DDXPlus selection and generation | In selection, whether the gold diagnosis appears in the model's top three ranked candidate options. In generation, whether the gold primary diagnosis appears among the three generated diagnoses after strict judge-based disease matching. |

| Metric | Used on | Operational definition in this study |
|---|---|---|
| Val. | ART and e-CARE generation | Percentage of outputs judged valid under the task-specific judge prompt, computed on the 400-example generation subset for each task. |
| WR-H / WR-C / WR-C+L | ART and UNcommonsense generation | Pairwise judge-based preference rates against a reference explanation: the human reference for ART (WR-H), crowd-written references for UNcommonsense (WR-C), and crowd-written references augmented with LLM-enhanced references for UNcommonsense (WR-C+L). The judge prompts also allow tie labels, which are tracked separately from outright wins. |
| CEQ | e-CARE generation | Causal Explanation Quality score, implemented to follow (Du et al., 2022) as closely as possible: for generated explanation $X$, $\text{CEQ}(X) = cs(C, E \mid X) - cs(C, E)$, where $cs(C, E \mid X) = \max[cs(C+X, E), cs(C, E+X)]$. In words, CEQ measures how much the explanation increases the estimated causal strength of the cause–effect pair. |
| Set-F1@3 | DDXPlus generation | Set-level F1 between the generated set of three diagnoses and the gold differential set (up to three diagnoses), after strict judge-based disease matching. Some cases contain fewer than three gold diagnoses when the benchmark differential collapses to a smaller set. |
| F1 | ProofWriter | Overlap-based F1 between the predicted set of valid single missing facts and the gold set. This complements exact-set accuracy by rewarding partially correct multi-answer outputs. |
| Tok-F1 | AbductionRules | Average token-level F1 against the gold missing fact. This is a secondary overlap metric added to distinguish near misses from fully incorrect generations. |
| CLS. | NeuLR | Mean character-level Levenshtein similarity between the generated missing fact and the gold missing fact. This is a secondary surface-form similarity metric added to capture near-correct outputs. |
| BLEU / ROUGE-L / BERTScore | ART, e-CARE, UNcommonsense generation | Reference-based lexical or semantic similarity metrics. They are reported as complementary signals only, since lexical similarity often underestimates explanation quality in many-to-one abductive settings. On UNcommonsense, BERTScore is computed against the best-matching human reference for each example. |

## Reading the Benchmark Tables

Several cautions matter when interpreting the results in Section 4.1. First, the suite is deliberately heterogeneous: it mixes short commonsense bridging, clinical differential diagnosis, long-context narrative inference, and formally constrained missing-premise generation. Second, several benchmarks are evaluated on fixed subsets or on only one official split. The most important examples are e-CARE, where we use the *cause* split only; MuSR, where we use only the murder-mysteries split; ProofWriter, where we use a depth-balanced D5-Ab subset; AbductionRules, where we use a stratified 400-example subset; and NeuLR, where we use 250 examples from the abductive split.

Third, several reported metrics are not native benchmark scores. DDXPlus generation is our own open-ended task formulation rather than an official benchmark task. Likewise, the judge-based metrics in ART, e-CARE, UNcommonsense, and DDXPlus generation are based on an LLM judge rather than on the human-evaluation procedures used in some source papers. This is especially important for cross-paper comparison: our results should be interpreted in light of the splits, prompts, task formulations, and evaluation metrics used in this study.

This heterogeneity is not a defect of the suite; it is part of the methodological point. Section 4.1 is not intended to compress abductive reasoning into one scalar score. Rather, it is meant to show how performance changes when the

abductive burden changes: generation versus selection, short versus long context, and naturalistic explanation versus structurally verifiable missing-premise completion.

## 9.2 Models, Methods, and Datasets for Comparative Analysis

This appendix provides a fine-grained breakdown of the datasets, models, and methods gathered from the three core papers (Sheng et al., 2025; Dougrez-Lewis et al., 2025; Xu et al., 2025a) analyzed in Section 4.2. It is important to emphasize that the evaluation space is sparse; not every model was tested with every method across all datasets. For exact experimental configurations and paired results, we refer readers to the original works.

Table 14: Datasets and corresponding reasoning types analyzed from the surveyed papers.

| Paper | Test Dataset | Reasoning Type |
|---|---|---|
| Sheng et al. (2025) | UniADILR-PSy (abduction split) | Abduction |
| Sheng et al. (2025) | UniADILR-PSy (deduction split) | Deduction |
| Sheng et al. (2025) | UniADILR-PSy (induction split) | Induction |
| Sheng et al. (2025) | UniADILR-HGc (abduction split) | Abduction |
| Sheng et al. (2025) | UniADILR-HGc (deduction split) | Deduction |
| Sheng et al. (2025) | UniADILR-HGc (induction split) | Induction |
| Dougrez-Lewis et al. (2025) | VitaminC (author-selected abductive instances) | Abduction |
| Dougrez-Lewis et al. (2025) | CLIMATE-FEVER (author-selected abductive instances) | Abduction |
| Dougrez-Lewis et al. (2025) | PHEMEPlus (author-selected abductive instances) | Abduction |
| Dougrez-Lewis et al. (2025) | VitaminC (author-selected deductive instances) | Deduction |
| Dougrez-Lewis et al. (2025) | CLIMATE-FEVER (author-selected deductive instances) | Deduction |
| Dougrez-Lewis et al. (2025) | PHEMEPlus (author-selected deductive instances) | Deduction |
| Xu et al. (2025a) | $\alpha$-NLI | Abduction |
| Xu et al. (2025a) | $\alpha$-NLG | Abduction |
| Xu et al. (2025a) | AbductiveRules | Abduction |
| Xu et al. (2025a) | D*Ab | Abduction |
| Xu et al. (2025a) | bAbI-15 | Deduction |
| Xu et al. (2025a) | EntailmentBank | Deduction |
| Xu et al. (2025a) | RuleTaker | Deduction |
| Xu et al. (2025a) | FOLIO | Deduction |
| Xu et al. (2025a) | Leap-Of-Thought | Deduction |
| Xu et al. (2025a) | bAbI-16 | Induction |
| Xu et al. (2025a) | CLUTRR | Induction |

Table 15: Models evaluated within each referenced paper.

| Paper | Evaluated Models |
|---|---|
| (Sheng et al., 2025) | LLaMA2-7B, LLaMA2-13B, LLaMA2-70B, GPT-3.5, GPT-4, T5-780M, T5-3B |
| (Dougrez-Lewis et al., 2025) | Claude V3 Sonnet, GPT-4, GPT-4o |
| (Xu et al., 2025a) | LLaMA3.1-Chat, Mistral-Ins-v0.3, Claude-3.5, GPT-4 |

Table 16: Inference and training methods utilized across the aggregated experiments.

| Method Category | Description |
| --- | --- |
| $k$-shot Prompting | In-context learning using $k \geq 0$ demonstration examples. |
| Chain-of-Thought (CoT) | Prompting models to generate intermediate reasoning steps. |
| Supervised Fine-Tuning (SFT) | Fine-tuning on specific datasets (not always matched to the test reasoning type). |

