# OpenReview forum: "Wiring the ‘Why’: A Unified Taxonomy and Survey of Abductive Reasoning in LLMs"
_TMLR — Under review for TMLR_

### Review · Reviewer_cBvY · 2026-05-26

**Summary Of Contributions:**

This paper presents what the authors describe as the first survey dedicated to abductive reasoning in LLMs. Its central contribution is a two-stage working definition (Hypothesis Generation followed by Hypothesis Selection) grounded in a careful reading of Peirce and Lipton, which is then used to organize a four-axis taxonomy (task formulation, dataset type, methodology, evaluation) over 60+ papers. The authors complement the review with a compact benchmark across 11 LLMs (3B to 72B) on nine datasets spanning both stages and both commonsense and expert/formal domains, plus a small cross-paradigm comparison aggregating prior results on abduction vs. deduction and induction. They close with identified gaps and concrete future directions.

Strengths

S1) The historical and philosophical grounding in Section 2.1 is unusually thorough for an ML/NLP survey, and it pays off: the two-stage definition emerges as a principled response to a genuine ambiguity in Peirce's own writing, rather than as an arbitrary engineering choice. The framework also does real work in Table 2, where it cleanly accommodates papers that had previously been talking past each other under the same label.

S2) Section 5.2's observation that adjacent fields (clinical reasoning, code debugging, culprit detection) often study what is essentially abduction under different labels is one of the more valuable framings in the paper. I would not be surprised to see it picked up and developed by others.

S3) The empirical section is also commendably honest about its scope. The authors flag subset choices, judge-based scoring, and the limits of within-suite comparisons in the main text. This is not the norm for surveys that include benchmarks.

Weaknesses

W1) My main concerns are with the empirical section and how the survey positions itself. The benchmark uses Gemini 3 Flash as the LLM judge for pairwise comparisons that include competing flagship models from other providers, and the implications of this choice are barely surfaced. Model identifiers like "GPT-5.4" appear in the tables without snapshot/access details, which makes reproduction difficult. The cross-paradigm comparison in Section 4.2 aggregates only three papers but draws a fairly confident conclusion about asymmetries between abductive and deductive performance.

W2) The "first survey" framing is also stronger than I think the evidence supports given prior coverage in LogiGLUE (Luo et al., 2023), He and Chen (2025), and Yang et al. (2023). All three are in the bibliography but none is contrasted in the body. A narrower claim is defensible, but the current one is not.

W3) A few smaller issues. Several taxonomy categories overlap in practice (M1 prompting and M4 multi-agent in particular), and the criteria for primary-category assignment in Table 2 are implicit. A number of 2026-dated references appear without indication of whether they are accepted, forthcoming, or preprints. The philosophical sections cite Aliseda (2006) and Dellsén (2024) as foundational but do not engage with what either author distinctively contributes, and well-known critiques of IBE are absent.

**Additional Comments:**

None

**Audience:**

Yes

**Audience Explanation:**

Abductive reasoning sits at the center of several application areas TMLR readers care about: scientific discovery, medical diagnosis, software debugging, investigative reasoning. The field has lacked a shared vocabulary, and this paper provides one. The benchmark is also a useful snapshot under a consistent prompting regime. I expect the two-stage framework in particular to be picked up and cited.

One accessibility caveat: the philosophical sections are dense by ML/NLP standards, and readers coming from purely empirical backgrounds may bounce off Section 2.1. A short reading guide would help. I do not consider this blocking.

**Broader Impact Concerns:**

The paper has no Broader Impact Statement. For a survey, this is partially defensible, but a brief one would be appropriate for two reasons. First, the paper repeatedly motivates abduction through high-stakes applications (medicine, law, scientific discovery), and Section 5.3's point that accuracy may not reflect genuine reasoning has obvious deployment implications that go unsaid. Second, the proposed future directions on circuit-level steering (Section 6.4) carry dual-use considerations: tools that bias models toward chosen explanatory framings could be misused in politically or ethically sensitive abductive tasks. A short paragraph acknowledging both points would be enough.

**Claims And Evidence:**

Yes

**Claims Explanation:**

The core contributions (the two-stage definition, the taxonomy, the gap analysis) are well supported through philosophical exposition and a careful structured literature review. The framework is convincingly motivated and demonstrably accommodates the heterogeneous prior literature in Table 2. The benchmark numbers are also reported carefully, with subset selection and methodological choices clearly flagged.

The reservations concern three specific places where the evidence is thinner than the framing suggests. First, the "first survey" claim should be contrasted with adjacent surveys rather than simply asserted. Second, the abduction vs. deduction asymmetry in Section 4.2 is plausible, but it rests on three aggregated papers and a sparse paired-experiment set; the framing should match the evidence. Third, the LLM-judge methodology is a load-bearing choice for several reported metrics (WR-H, WR-C, WR-C+L, validity scores) and deserves either a defense or a small calibration study, not just an appendix mention.

**Requested Changes:**

Critical changes:

C1. Defend or audit the LLM-judge methodology. Gemini 3 Flash (preview) is used to score pairwise generation comparisons across competing flagship models from other providers. Please add either (a) a small human-judged calibration sample on one open-ended task, (b) a second independent judge model on a subset, or (c) at minimum a prominent discussion of this choice in Section 4.1 rather than only in the appendix. Several of the reference-free metrics in Tables 4 and 5 depend heavily on this single judge, so the current presentation understates a load-bearing methodological choice.

C2. Make the benchmark reproducible. Tables 3 through 6 reference identifiers like "GPT-5.4" and "DeepSeek-V3.2" without snapshot version, release date, or access method. Please specify these for each model. Also, Section 4.1 and Appendix 8.1 note that several benchmarks use restricted subsets (500 ART examples, 400 DDXPlus examples, 250 MuSR cases, etc.) but do not specify how those subsets were drawn or whether the example IDs will be released. Please address both.

C3. Reframe the "first survey" claim. Add a brief related-work paragraph in Section 1 contrasting this survey with LogiGLUE (Luo et al., 2023), He and Chen (2025), and Yang et al. (2023). The narrower claim, "the first survey dedicated specifically to abductive reasoning in LLMs," is defensible and should be made explicitly, with acknowledgment of overlapping coverage in adjacent surveys.

C4. Tighten Section 4.2 or strengthen its evidence. The cross-paradigm comparison aggregates only three papers, and most existing benchmarks isolate a single reasoning paradigm. Please (a) expand the discussion of what cannot be concluded from this analysis, (b) note in the captions of Figures 7 and 8 that these are aggregated observational comparisons rather than controlled experiments, and ideally (c) run a small controlled follow-up on a subset of the benchmark models for a within-paper apples-to-apples comparison. The conclusion is interesting, but its current framing outruns the evidence.

Recommended changes:

R1. Discuss model-level reversals. Tables 3 through 5 contain interesting non-monotonicities (GPT-4o leading GPT-5.4 on MuSR, DeepSeek-V3.2 underperforming smaller open-weight models on some metrics, Qwen3 32B sometimes beating Qwen2.5 72B). A targeted discussion would be more informative than the current "benchmark-dependent" framing.

R2. Engage with reward hacking in Section 6.1. The proposal to use RL to align models with abstract explanatory virtues should engage with the recent literature on reward hacking in RLHF/RLAIF. A short paragraph on how virtue-based rewards might be gamed, and how to guard against this, would strengthen the proposal considerably.

R3. Minor edits. Abstract: "output structure.." has a double period. The choice of judge model should appear in the main text, not only in the appendix. The bibliography is inconsistent in how preprints are formatted (some entries include arXiv IDs, others do not).

---

> ### Author Response · Authors · 2026-06-13
> **Response to Reviewer cBvY (1/2)**
>
> Due to the OpenReview comment limit, we split our response into two parts.
>
> ### Accessibility / reading guide
>
> We thank the reviewer for this helpful accessibility suggestion. We agree that Section 2.1 is conceptually denser than the later survey and benchmarking sections, especially for readers coming from a primarily empirical ML/NLP background. To address this, we added a short Guide for the Reader paragraph at the end of the Introduction, immediately before Section 2. The new paragraph explains why the historical discussion is included, clarifies that it motivates the unified two-stage framework introduced in Section 2.2.1, and notes that readers mainly interested in the operational framework and empirical analysis can begin there and return to Section 2.1 as needed.
>
> We believe this addition makes the paper easier to navigate without reducing the importance of the historical and conceptual foundations.
>
> ### W1/C1. LLM-judge methodology
>
> We agree that the judge choice is a load-bearing methodological decision for the reference-free Stage I metrics. To address this, we added an explicit LLM-as-a-Judge Audit to Section 4.1. Concretely, we re-scored 50 sampled examples each from ART, DDXPlus, and UNcommonsense with an independent judge, Claude Haiku 4.5, using the same evaluation prompts. This yields 550 paired judge decisions per dataset.
>
> The main pooled results are now reported in Table 7:
>
> - ART validity: 83.3 (Gemini) vs. 76.7 (Claude), agreement 87.3, kappa 0.60
> - DDXPlus Hit@3: 49.1 vs. 50.4, agreement 93.6, kappa 0.87
> - UNcommonsense WR-C+L: 25.6 vs. 44.2, agreement 67.8, kappa 0.32
>
> These results support the main comparative trends while also making clear that absolute reference-free scores -- especially on open-ended preference tasks -- are judge-dependent estimates rather than judge-independent ground truth.
>
> ### W1/C2. Benchmark reproducibility
>
> We added a new appendix subsection, Model Snapshots and Restricted Subsets, in Appendix 9.1. Table 12 now reports the exact model identifier, snapshot/release information, and access method used for each model in Tables 3--6.
>
> We also clarified how restricted subsets were constructed. Specifically, subsets were drawn once before evaluation by uniform sampling without replacement from the stated official split, except where Tables 10--11 specify an explicit stratification constraint. We have uploaded the sampled subsets as anonymized supplementary material for review. Appendix 9.1 currently includes a repository placeholder, which will be replaced with the final public repository link after review.
>
> ### W1/C4. Cross-paradigm comparison
>
> We agree that Section 4.2 is best framed as an observational synthesis rather than a controlled experiment, and we have revised the manuscript to reflect this.
>
> Regarding the suggestion for a controlled, within-paper experiment (point c), we agree this would be highly valuable. However, due to resource constraints, we were unable to conduct new systematic evaluations on a representative subset of these models for this revision.
>
> Instead, we have focused on addressing points (a) and (b) by clarifying the observational nature of the analysis and tempering our claims. Specifically, we made the following updates to Section 4.2:
>
> 1. Clarified Figure Captions (Addressing b): We updated the captions of Figure 7 and Figure 8 to explicitly designate them as "aggregated observational comparisons" compiled from external studies, rather than controlled, uniform experiments.
> 2. Expanded Discussion of Limitations (Addressing a): We revised the final paragraph of Section 4.2 to detail the limitations of this observational data. We now explicitly state that we do not draw causal conclusions, do not assume exact semantic or structural alignment between the paired tasks, and do not claim universal generalization from the three source papers.
> 3. Tempered Textual Claims: We adjusted the phrasing throughout Section 4.2 to ensure the discussion remains aligned with the observational evidence.
>
> ### W2/C3. First-survey claim
>
> We agree that the original novelty claim was too broad. In the revised manuscript, we now state the narrower claim that this is, to the best of our knowledge, the first survey dedicated specifically to abductive reasoning in LLMs.
>
> To make that positioning explicit, we revised the Introduction's discussion of prior surveys to situate our paper relative to Luo et al. (2023), Yang et al. (2023), and He and Chen (2025), while acknowledging the overlap. We also clarify the distinguishing focus of our paper: abductive reasoning itself as the central object of study, a unified two-stage working definition, an abductive-specific four-axis taxonomy, and a compact empirical benchmark spanning both stages.

---

> > ### Comment · Reviewer_cBvY · 2026-07-10
> > **thank you**
> >
> > Thank you for addressing my comments.

---

> ### Author Response · Authors · 2026-06-13
> **Response to Reviewer cBvY (2/2)**
>
> ### W3. Taxonomy overlap, 2026 references, philosophical citations, and IBE critiques
>
> Thank you for highlighting these points.
>
> - Taxonomy overlap / M1 vs. M4. We now clarify this directly in Section 3.3. While multi-agent systems often use prompt-based instructions internally, we distinguish M4 by its explicit distribution of roles and iterative communication among components, whereas M1 concerns steering a single model call.
> - Assignment criteria in Table 2. To address taxonomic overlaps and clarify our criteria for Table 2, we added brief guidelines to the introductory paragraphs of Sections 3.1-3.4. Generally, papers are classified under their dominant characteristic. For papers that present or evaluate multiple independent components (such as proposing both a fine-tuning and a prompting method, or conducting separate human and automatic evaluations), we list all applicable categories.
> - 2026 references. We reviewed the bibliography so that recent references are labeled consistently as preprints, accepted papers, or forthcoming venue publications where applicable.
> - Aliseda / Dellsen. We acknowledge that our citation of Aliseda (2006) and Dellsén (2024) does not engage with their specific philosophical theses. As this survey is designed for the computer science and large language model (LLM) community, we reference these works strictly as historical resources to construct a descriptive timeline of how abductive reasoning has evolved. This approach allows us to connect historical shifts to current definitional inconsistencies in LLMs without diverting focus from our primary objective of developing a computational framework. To make this intent clear to the reader, we have updated Section 2.1 of the manuscript. The revised text explicitly states that these works are cited for their historical syntheses rather than to engage with their individual philosophical arguments.
> - Critiques of IBE. The omission of classic philosophical critiques of Inference to the Best Explanation (IBE) is an intentional scoping decision. The objective of this survey is not to evaluate the epistemological validity of IBE, but rather to unify the fragmented LLM research landscape. Discussing these philosophical debates would diverge from this computational focus, which is aimed at establishing clear, empirical evaluation frameworks. Instead, IBE is introduced strictly for structural and terminological utility. Structurally, Peter Lipton's two-stage framework closely mirrors how abductive reasoning is operationalized in LLM pipelines through generation and selection stages. Terminologically, referencing IBE helps reconcile differences in recent LLM literature, where some papers use "IBE" and others use "abductive reasoning" to describe virtually identical computational tasks, thereby demonstrating that these studies share a common methodological foundation.
>
> ### R1. Model-level reversals
>
> We appreciate this helpful suggestion. We agree that these model-level crossovers are interesting. We already note this pattern in the “Scale and Model Family Effects” paragraph of §4.1: although performance often improves with scale within a family, rankings across leading models are not monotonic. In our suite, for example, DeepSeek-V3.2 does not consistently outperform the strongest Qwen/Llama models, and GPT-4o exceeds GPT-5.4 on some tasks, including MuSR. We believe the current §4.1 discussion already captures this point, so rather than adding a separate paragraph, we clarified the interpretation here. More broadly, these reversals are not unusual; benchmark- and subskill-specific crossovers are common in current reasoning evaluations.
>
> ### R2. Reward hacking
>
> We substantially revised Section 6.1 to engage with reward hacking in RLHF/RLAIF-style optimization. The new text now discusses how explanatory virtues can be gamed superficially -- for example, parsimony collapsing into underspecified hypotheses and coherence degenerating into fluent but circular explanations.
>
> We also added mitigation strategies drawn from the alignment literature, including step-level process rewards, multi-signal objectives, reference-policy regularization, adversarial reward-model training, and causal reward modeling.
>
> ### R3. Minor edits
>
> We addressed the requested minor issues:
>
> 1. The abstract typo after "output structure" was corrected.
> 2. The judge model is now identified in the main text (Section 4.1), not only in the appendix.
> 3. Preprint formatting was standardized across the bibliography.
>
> ### Broader Impact Statement
>
> We added a Broader Impact Statement as the final section of the paper. It now explicitly discusses:
>
> - The deployment risk of treating benchmark accuracy as evidence of genuine abductive reasoning in high-stakes domains such as medicine, law, and scientific discovery.
> - The dual-use risk of circuit-level steering or interpretability tools being used to bias models toward chosen explanatory framings in sensitive contexts.

---

### Review · Reviewer_bPdw · 2026-06-01

**Summary Of Contributions:**

This paper provides a combination of introductory text on abductive reasoning and evaluated many open-source and closed-source LLMs on abductive reasoning tasks.

**Audience:**

No

**Audience Explanation:**

I admit that different reasoning paradigms are of interests to people in NLP, linguistics, philosophy (logic), and psychology.
I personally lack the ability to fully judge if this paper provided fresh/novel/new insights from the previous papers. The paper appears to be an amalgamation of college freshman introduction to abductive reasoning text and some preliminary analysis of LLM results from different papers.

**Broader Impact Concerns:**

Although I generally support slightly niche subjects being accepted to TMLR, and logic for sure is one of such subjects, this paper doesn't seem to offer interesting insights beyond being an introductory textbook of sorts.

Maybe this is what TMLR survey paper is looking for -- I'm not so sure about what the acceptance threshold is, so I'll leave it to other reviewers and Action Editor to decide if this is of interests to more people!

**Claims And Evidence:**

Yes

**Claims Explanation:**

- The paper did provide a unified definitional framework
- The paper classified existing benchmarks into their framework and re-analyzed LLM's performance in stage 1 and stage 2.
- The paper provided very preliminary/simple analysis over accuracy distributions over reasoning types

**Requested Changes:**

I think it would be nice if a few things are enhanced:
1. What new capabilities should we require LLMs to have/do? GPT-5.4 can achieve 72%-76% already, what is to say our current training paradigm won't get us to 90% or even 100%?
2. Does the framework you proposed yield new insights of what LLMs are currently lacking? Stage 1/2 LLM numbers again, look similar (72 vs 76).

Out of curiosity, if you analyze an LLM's chain-of-thought -- you can do that with the Qwen models, are you able to identify accurately if a CoT is using abductive reasoning or inductive reasoning or any of the types you mentioned? Do you think it brings value to the LLM community that LLM uses the **right** type of reasoning or do you think succeeding in this task will improve LLM's reasoning capabilities overall?

---

> ### Author Response · Authors · 2026-06-13
> **Response to Reviewer bPdw**
>
> We thank the reviewer for their valuable feedback.
>
> ### What new capabilities should we require LLMs to have/do?
>
> We do not claim that current training paradigms cannot continue improving benchmark scores, including much higher scores on some tasks. Our point is different: because abductive reasoning is currently studied through fragmented task formulations and evaluation setups, it is often unclear what progress actually means across papers. The capability we want to make more visible is not just higher accuracy on static benchmarks, but stronger explanatory reasoning: generating plausible hypotheses from incomplete observations, comparing competing explanations, and doing so in ways that remain useful in richer, uncertainty-heavy settings. In that sense, the value of our framework is to help organize the field so that future improvements—whether they come from current training paradigms or new ones—can be interpreted more coherently.
>
> ### Does the proposed framework yield new insights into what LLMs are currently lacking?
>
> Yes, but not through a simple comparison of overall Stage I and Stage II averages. Those aggregates combine tasks with different structures, metrics, and difficulty profiles, so numbers like “72 vs. 76” are not, by themselves, a meaningful diagnosis of which stage is harder. The more informative comparison is on paired tasks within the same domain. In our suite, DDXPlus gives the clearest example: performance is much stronger when the model ranks candidate diagnoses than when it must generate diagnoses from scratch. More broadly, the framework helps separate several kinds of difficulty that are often conflated under a single “abduction” label—generation vs. selection, short vs. long context, open-ended explanation vs. structurally constrained completion, and metric sensitivity. We therefore see the main insight of the framework as clarifying where performance differences come from, rather than making a strong global claim that one stage is always weaker than the other. A more definitive diagnosis of stage-specific weaknesses would require carefully matched experimental design, which we view as an important direction for future work.
>
> ### Chain-of-thought reasoning-type analysis and the value of abduction
>
> We appreciate this thoughtful question. Even for models such as Qwen that expose reasoning traces, our view is that it may be possible in principle to inspect a visible CoT and infer what kind of reasoning it is using, but not yet in a way we would treat as fully reliable. For example, a trace looks abductive when it starts from observations, proposes candidate explanations that would make them unsurprising, and then compares those explanations by plausibility; it looks deductive when it mainly applies rules to derive what must follow; and it looks inductive when it generalizes from repeated cases or patterns. So we do think this kind of CoT analysis could be useful. At the same time, exposed CoT is prompt-sensitive, noisy, and may not faithfully reflect the model’s actual internal process, so we would currently treat such analysis as suggestive rather than definitive. A more rigorous answer would require dedicated criteria, annotation protocols, and controlled experiments, which we see as an important future direction.
>
> There is value in knowing whether an LLM is using the intended type of reasoning, because that helps us diagnose what the model is actually doing and whether benchmark success reflects the capability we care about.

---

### Review · Reviewer_Ggcq · 2026-06-02

**Summary Of Contributions:**

This paper presents a comprehensive survey and taxonomy of abductive reasoning (the process of inferring the most plausible explanation for an observation) in LLMs. To address the field's widespread conceptual fragmentation, the authors establish a two-stage definition that disentangles the reasoning process into Hypothesis Generation (creating candidate explanations to bridge epistemic gaps) and Hypothesis Selection (evaluating candidates to choose the most plausible explanation).

Using this foundation, they categorize existing literature across a four-axis taxonomy based on task formulation, dataset type, methodology, and evaluation approach. Furthermore, through an empirical benchmarking of modern LLMs (ranging from 3B to 72B parameters), the study reveals that while models perform relatively well on static hypothesis selection tasks, they struggle significantly with open-ended hypothesis generation and complex narrative or domain constraints. Finally, the paper identifies critical gaps in current research (such as shallow benchmark design and limited mechanistic understanding) and advocates for future directions, including RL optimized for explanatory virtues, multi-agent frameworks, and advanced circuit-level interpretability.

## Key Strengths
- The two-stage definition of abductive reasoning (Hypothesis Generation and Hypothesis Selection) is pretty clear, and it resolves conceptual fragmentation in the field.

- The authors introduce a novel, four-axis taxonomy (categorizing task formulation, dataset type, methodology, and evaluation approach) to systematically organize and review over 60 prior works.

- It grounds its theoretical framework with a practical empirical evaluation, testing a wide range of modern LLMs (from 3B to 72B parameters) across both commonsense and expert domains.

## Key Weaknesses
- The authors rely on an LLM to evaluate the open-ended generative tasks, but they specifically chose Gemini 1.5 Flash (preview) as their judge. Using a smaller model to evaluate the abductive reasoning quality of frontier models (like GPT-4o, DeepSeek-V3.2, or Llama 3.3 70B) could be a red flag. Flash may lack the reasoning depth to reliably critique complex hypothesis generation.


- The study explicitly states that all experiments were run using deterministic decoding (temperature = 0). While fine for multiple-choice selection, greedy decoding severely handcuffs models during Stage I (Hypothesis Generation). Abductive reasoning is inherently about exploring a diverse space of plausible explanations; testing generation at temperature 0 kills the creativity the authors are trying to measure.


- Rather than running full, standardized test sets, the authors used heavily restricted subsets (e.g., 200 examples for DDXPlus, 300 for ART generation) and created their own custom task formulations. This degrades the statistical significance of their findings and makes it impossible to directly compare their numbers with established SOTA baselines.


- The authors strongly criticize the field for relying on "static, single-shot benchmarks" that fail to capture the true, evolving nature of abductive reasoning. Yet, their own empirical contribution relies entirely on static, single-shot, zero-shot direct instruction prompts held fixed across models. They didn't even systematically test Chain-of-Thought prompting in their core suite to see if intermediate reasoning steps improve hypothesis generation.

**Audience:**

Yes

**Audience Explanation:**

This paper will be highly relevant to AI researchers who focused on advancing the reasoning capabilities of LLMs.

**Claims And Evidence:**

Yes

**Claims Explanation:**

The support for the claims in this paper is a bit mixed. It is highly convincing on the theoretical front but relies on flawed evidence for its empirical conclusions.

- Theoretical Claims: The authors' primary claim that the field of abductive reasoning is conceptually fragmented and reliant on shallow benchmarks is very well-supported. The evidence here is qualitative but undeniable, backed by a thorough literature review and a rigorous taxonomy that maps the disjointed approaches of over 60 different papers.

- Empirical Claims on Model Capabilities (Weakly Supported): Claims regarding actual LLM performance (such as the assertion that models struggle significantly more with generation than selection) are built on a shaky experimental foundation. The evidence is compromised because the authors used suboptimal greedy decoding (temperature = 0) for creative generation tasks, tested on heavily restricted data subsets rather than full benchmarks, and used a smaller model (Gemini Flash) to evaluate the outputs of frontier models.

**Requested Changes:**

See "Key Weaknesses"

---

> ### Author Response · Authors · 2026-06-13
> **Response to Reviewer Ggcq**
>
> We thank the reviewer for their meticulous reading of the paper and for their nuanced feedback.
>
> ### Key Weakness 1. LLM-as-judge concern
>
> Thank you for raising this concern. First, to correct a factual point: the judge used in our experiments is Gemini 3 Flash (preview), not Gemini 1.5 Flash. We now state this explicitly in the main text of Section 4.1, in addition to Appendix 9.1.
>
> More importantly, we also added a direct judge-model audit (Table 7) to address the broader reliability concern. Using Claude Haiku 4.5 as an independent judge on sampled subsets of ART, DDXPlus, and UNcommonsense, we find that judge sensitivity varies by task: DDXPlus is quite stable across judges, ART shows solid agreement with some strictness differences, and UNcommonsense is clearly the most judge-sensitive. We have revised the manuscript accordingly so that these metrics are presented with the appropriate caution.
>
> ### Key Weakness 2. Deterministic decoding / temperature
>
> We agree that Stage I generation may be sensitive to decoding choices. To test this directly, we added a temperature ablation (Table 8) on 100-example subsets of DDXPlus, UNcommonsense, and ProofWriter for Qwen3 32B, Llama3.3 70B, and GPT-5.4, varying only the decoding temperature from T=0 to T=0.7 under the same direct-output prompt.
>
> The results are mixed rather than uniformly positive. For example, DDXPlus improves slightly for the two open-weight models, while UNcommonsense and ProofWriter do not show a consistent gain. We therefore revised the paper to state that the main Stage I trends do not appear to be merely artifacts of greedy decoding, although open-ended abductive generation remains sensitive to decoding and evaluation choices.
>
> ### Key Weakness 3. Restricted subsets and task formulations
>
> We agree that small restricted subsets limit statistical strength and complicate direct comparison to prior work. To improve this, we increased several Stage I evaluation sizes in the revised manuscript: ART to 400 examples, e-CARE to 400, DDXPlus to 400, and AbductionRules to 400.
>
> We also document the subset-construction procedure and task formulations in Appendix 9.1 (Tables 10--12). For DDXPlus generation in particular, we make explicit that this is our open-ended formulation rather than an official benchmark task, and we caution readers accordingly when comparing against prior results.
>
> ### Key Weakness 4. Static single-shot prompting and Chain-of-Thought
>
> We agree that this concern should be tested empirically. To address it, we added a CoT ablation (Table 8) on the same 100-example Stage I subsets. This shows that explicit intermediate reasoning can help on more structured tasks such as ProofWriter and, for some models, DDXPlus -- but it does not uniformly improve open-ended abductive generation, and it substantially hurts UNcommonsense for some models.
>
> In addition, to complement the fixed-prompt benchmark suite, we added a small interactive SPLAT diagnostic (Table 9). This multi-turn setting is useful because the model must iteratively ask questions, update a provisional explanation, and revise its hypothesis as new evidence arrives. The results remain far from saturated, especially on harder puzzles, which supports our broader argument that dynamic abductive settings are still difficult for current LLMs.

---

### Author Response · Authors · 2026-06-13
**General Response and Summary of Revisions**

Dear Reviewers and Action Editor,

We thank the reviewers for their careful reading of the paper and for the constructive feedback. In response, we substantially revised the manuscript. The main changes are:

- Section 4.1 now includes an explicit LLM-as-a-Judge Audit (Table 7), plus prompting/decoding ablations (Table 8) and a small interactive SPLAT diagnostic (Table 9).
- Appendix 9.1 now includes a new Model Snapshots and Restricted Subsets subsection (Table 12) documenting model identifiers, access methods, and subset-construction details.
- The Introduction now narrows the novelty claim to "the first survey dedicated specifically to abductive reasoning in LLMs" and revises the discussion of prior surveys to position our paper relative to related survey work.
- The Introduction also now includes a short Guide for the Reader paragraph to improve accessibility and help readers primarily interested in the operational framework and empirical analysis navigate Section 2.
- Sections 3.1--3.4 now make the taxonomy-assignment criteria explicit, and Section 3.3 now clarifies the distinction between M1 prompting and M4 multi-agent frameworks.
- Section 4.2 has been reframed more cautiously: Figures 7 and 8 are now explicitly described as aggregated observational comparisons, and the text now states what cannot be concluded from this analysis.
- Section 6.1 now explicitly discusses reward hacking / specification gaming and corresponding safeguards.
- A Broader Impact Statement has been added.
- We also corrected minor issues in the abstract and bibliography formatting and moved the judge-model description into the main text.

We reply separately to each reviewer in the corresponding discussion thread below.